# Eliminating Domain Bias for Federated Learning in Representation Space

**Jianqing Zhang**[1], **Yang Hua**[2], **Jian Cao**[1]*, **Hao Wang**[3],
**Tao Song**[1] **Zhengui Xue**[1], **Ruhui Ma**[1]*, **Haibing Guan**[1]

[1]Shanghai Jiao Tong University   [2]Queen's University Belfast   [3]Louisiana State University

{tsingz, cao-jian, songt333, zhenguixue, ruhuima, hbguan}@sjtu.edu.cn
Y.Hua@qub.ac.uk, haowang@lsu.edu

## Abstract

Recently, federated learning (FL) is popular for its privacy-preserving and collaborative learning abilities. However, under statistically heterogeneous scenarios, we observe that biased data domains on clients cause a *representation bias* phenomenon and further degenerate generic representations during local training, *i.e.*, the *representation degeneration* phenomenon. To address these issues, we propose a general framework ***Domain Bias Eliminator*** (DBE) for FL. Our theoretical analysis reveals that DBE can promote bi-directional knowledge transfer between server and client, as it reduces the domain discrepancy between server and client in representation space. Besides, extensive experiments on four datasets show that DBE can greatly improve existing FL methods in both generalization and personalization abilities. The DBE-equipped FL method can outperform ten state-of-the-art personalized FL methods by a large margin. Our code is public at https://github.com/TsingZ0/DBE.

## 1 Introduction

As a popular distributed machine learning paradigm with excellent privacy-preserving and collaborative learning abilities, federated learning (FL) trains models among clients with their private data kept locally [37, 56, 79]. Traditional FL (*e.g.*, the famous FedAvg [56]) learns one single global model in an iterative manner by locally training models on clients and aggregating client models on the server. However, it suffers an accuracy decrease under statistically heterogeneous scenarios, which are common scenarios in practice [47, 56, 67, 86].

Due to statistical heterogeneity, the data domain on each client is biased, which does not contain the data of all labels [37, 45, 67, 69, 79, 84]. As the received global model is locally trained on individual clients' biased data domain, we observe that this model extracts biased (*i.e.*, forming client-specific clusters) representations during local training. We call this phenomenon "*representation bias*" and visualize it in Figure 1. Meanwhile, by training the received global model with missing labels, the generic representation quality over all labels also decreases during local training [45]. Furthermore, we ob-

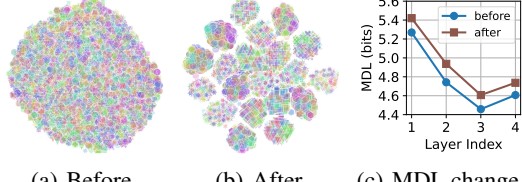

(a) Before.  (b) After.  (c) MDL change.

Figure 1: t-SNE [73] visualization and per-layer MDL (bits) for representations before/after local training in FedAvg. We use *color* and *shape* to distinguish *labels* and *clients* respectively for t-SNE. A large MDL means low representation quality. *Best viewed in color and zoom-in.*

---

*Corresponding authors.

serve that this "*representation degeneration*" phenomenon happens at every layer, as shown in Figure 1(c). We estimate the representation quality via minimum description length (MDL) [62, 65, 74], a metric independent of data and models, measuring the difficulty of classifying target labels according to given representations.

To tackle the statistical heterogeneity, unlike traditional FL that learns a single global model, personalized FL (pFL) comes along by learning personalized models (or modules) for each client besides learning a global model among clients [20, 22, 69]. Typically, most of the existing pFL methods train a personalized classifier[2] for each client [3, 14, 20, 61], but the feature extractor still extracts all the information from the biased local data domain, leading to representation bias and representation degeneration during local training.

To address the representation bias and representation degeneration issues in FL, we propose a general framework ***Domain Bias Eliminator*** (DBE) for FL including two modules introduced as follows. Firstly, we detach the representation bias from original representations and preserve it in a ***Personalized Representation Bias Memory*** (PRBM) on each client. Secondly, we devise a ***Mean Regularization*** (MR) that explicitly guides local feature extractors to extract representations with a consensual global mean during local training to let the local feature extractor focus on the remaining unbiased information and improve the generic representation quality. In this way, we turn one level of representation between the feature extractor and the classifier on each client into two levels of representation with a client-specific bias and a client-invariant mean, respectively. Thus, we can eliminate the *conflict* of extracting representations with client-specific biases for clients' requirements while extracting representations with client-invariant features for the server's requirements in the same representation space. Our theoretical analysis shows that DBE can promote the bi-directional knowledge transfer between server and client with lower generalization bounds.

We conduct extensive experiments in computer vision (CV) and natural language processing (NLP) fields on various aspects to study the characteristics and effectiveness of DBE. In both generalization ability (measured by MDL) and personalization ability (measured by accuracy), DBE can promote the fundamental FedAvg as well as other representative FL methods. Furthermore, we compare the representative FedAvg+DBE with ten state-of-the-art (SOTA) pFL methods in various scenarios and show its superiority over these pFL methods. To sum up, our contributions are:

- We observe the representation bias and per-layer representation degeneration phenomena during local training in the representative FL method FedAvg.

- We propose a framework DBE to memorize representation bias on each client to address the representation bias issue and explicitly guide local feature extractors to generate representations with a universal mean for higher generic representation quality.

- We provide theoretical analysis and derive lower generalization bounds of the global and local feature extractors to show that DBE can facilitate bi-directional knowledge transfer between server and client in each iteration.

- We show that DBE can improve other representative traditional FL methods including FedAvg at most **-22.35%** in MDL (bits) and **+32.30** in accuracy (%), respectively. Furthermore, FedAvg+DBE outperforms SOTA pFL methods by up to **+11.36** in accuracy (%).

## 2 Related Work

Traditional FL methods that focus on improving accuracy under statistically heterogeneous scenarios based on FedAvg including four categories: update-correction-based FL [25, 38, 60], regularization-based FL [1, 17, 40, 46], model-split-based FL [35, 45], and knowledge-distillation-based FL [27, 33, 88, 96]. For pFL methods, we consider four categories: meta-learning-based pFL [13, 22], regularization-based pFL [47, 67], personalized-aggregation-based pFL [21, 52, 87, 89], and model-split-based pFL [3, 14, 20, 61, 85]. Due to limited space, we only introduce the FL methods that are close to ours and leave the *extended version of this section* to Appendix A.

**Traditional FL methods.** MOON [45] utilizes contrastive learning to correct the local training of each client, but this input-wise contrastive learning still relies on the biased local data domain, so it still suffers from representation skew. Although FedGen [96] learns a shared generator on the server

---

[2]A model is split into a feature extractor and a classifier. They are sequentially jointed.

and reduces the heterogeneity among clients with the generated representations through knowledge distillation, it only considers the local-to-global knowledge for the single global model learning. On the other hand, FedGen additionally brings non-negligible communication and computation overhead for learning and transmitting the generator.

**pFL methods.** FedPer [3] and FedRep [20] keep the classifier locally, but the feature extractor still learns biased features without explicit guidance. Besides, their local feature extractors are trained to cater to personalized classifiers thus losing generality. FedRoD [14] reduces the discrepancy of local training tasks among clients by using a balanced softmax (BSM) loss function [64], but the BSM is useless for missing labels on each client while label missing is a common situation in statistically heterogeneous scenarios [50, 86, 89]. Moreover, the uniform label distribution modified by the BSM cannot reflect the original distribution. Differently, FedBABU [61] trains a global feature extractor with a naive and frozen classifier, then it fine-tunes the classifier for each client to finally obtain personalized models. However, the post-FL fine-tuning study is beyond the FL scope, as almost all the FL methods have multiple fine-tuning variants, *e.g.*, fine-tuning the whole model or only a part of the model. Like FedAvg, FedBABU still locally extracts biased features during the FL process.

## 3 Notations and Preliminaries

### 3.1 Notations

In this work, we discuss the statistically heterogeneous scenario in typical multi-class classification tasks for FL, where $N$ clients share the same model structure. Here, we denote notations following FedGen [96] and FedRep [20]. The client $i, i \in [N]$, has its own private data domain $\mathcal{D}_i$, where the data are sampled from $\mathcal{D}_i$. All the clients collaborate to train a global model $g$ parameterized by $\boldsymbol{\theta}$ without sharing their private local data.

Since we focus on representation learning in FL, we regard $g$ as the sequential combination of a feature extractor $f$ that maps from the input space $\mathcal{X}$ to a representation space $\mathcal{Z}$, *i.e.*, $f : \mathcal{X} \mapsto \mathcal{Z}$ parameterized by $\boldsymbol{\theta}^f$ and a classifier $h$ that maps from the representation space to the output space $\triangle^{\mathcal{Y}}$, *i.e.*, $h : \mathcal{Z} \mapsto \triangle^{\mathcal{Y}}$ parameterized by $\boldsymbol{\theta}^h$. Formally, we have $g := h \circ f$, $\boldsymbol{\theta} := [\boldsymbol{\theta}^f; \boldsymbol{\theta}^h]$, $\mathcal{X} \subset \mathbb{R}^D$ and $\mathcal{Z} \subset \mathbb{R}^K$. $\triangle^{\mathcal{Y}}$ is the simplex over label space $\mathcal{Y} \subset \mathbb{R}$. With any input $\boldsymbol{x} \in \mathcal{X}$, we obtain the feature representation by $\boldsymbol{z} = f(\boldsymbol{x}; \boldsymbol{\theta}^f) \in \mathcal{Z}$.

### 3.2 Traditional Federated Learning

With the collaboration of $N$ clients, the objective of traditional FL, *e.g.*, FedAvg [56], is to iteratively learn a global model that minimizes its loss on each local data domain:

$$\min_{\boldsymbol{\theta}} \; \mathbb{E}_{i \in [N]}[\mathcal{L}_{\mathcal{D}_i}(\boldsymbol{\theta})], \tag{1}$$

$$\mathcal{L}_{\mathcal{D}_i}(\boldsymbol{\theta}) := \mathbb{E}_{(\boldsymbol{x}_i, y_i) \sim \mathcal{D}_i}[\ell(g(\boldsymbol{x}_i; \boldsymbol{\theta}), y_i)] = \mathbb{E}_{(\boldsymbol{x}_i, y_i) \sim \mathcal{D}_i}[\ell(h(f(\boldsymbol{x}_i; \boldsymbol{\theta}^f); \boldsymbol{\theta}^h), y_i)], \tag{2}$$

where $\ell : \triangle^{\mathcal{Y}} \times \mathcal{Y} \mapsto \mathbb{R}$ is a non-negative and convex loss function. Following FedGen, we assume that all clients share an identical loss function $\ell$ and a *virtual* global data domain $\mathcal{D}$, which is the union of all local domains: $\mathcal{D} := \bigcup_{i=1}^{N} \mathcal{D}_i$. In practice, traditional FL methods [45, 46, 56] optimize Eq. (1) by $\min_{\boldsymbol{\theta}} \sum_{i=1}^{N} \frac{n_i}{n} \mathcal{L}_{\hat{\mathcal{D}}_i}(\boldsymbol{\theta})$, where $\hat{\mathcal{D}}_i$ is an observable dataset, $n_i = |\hat{\mathcal{D}}_i|$ is its size, and $n = \sum_{i=1}^{N} n_i$.

In each communication iteration, clients conduct local updates on their private data to train the global model $\boldsymbol{\theta}$ by minimizing their local loss. Formally, for client $i$, the objective during local training is $\min_{\boldsymbol{\theta}} \; \mathcal{L}_{\mathcal{D}_i}(\boldsymbol{\theta})$. The empirical version of $\mathcal{L}_{\mathcal{D}_i}(\boldsymbol{\theta})$ is $\mathcal{L}_{\hat{\mathcal{D}}_i}(\boldsymbol{\theta}) := \frac{1}{n_i} \sum_{j=1}^{n_i} \ell(h(f(\boldsymbol{x}_{ij}; \boldsymbol{\theta}^f); \boldsymbol{\theta}^h), y_{ij})$, which is optimized by stochastic gradient descent (SGD) [56, 90] in FedAvg.

## 4 Method

### 4.1 Problem Statement

pFL iteratively learns a personalized model or module for each client with the assistance of the global model parameters from the server. Our objective is (with a slight reuse of the notation $\mathcal{L}_{\mathcal{D}_i}$)

$$\min_{\boldsymbol{\theta}_1, \dots, \boldsymbol{\theta}_N} \; \mathbb{E}_{i \in [N]}[\mathcal{L}_{\mathcal{D}_i}(\boldsymbol{\theta}_i)], \tag{3}$$

where $\boldsymbol{\theta}_i$ is a model consisting of global and personalized modules. The global modules are locally trained on clients and shared with the server for aggregation like traditional FL, but the personalized modules are preserved locally on clients. Following traditional FL, we empirically optimize Eq. (3) by $\min_{\boldsymbol{\theta}_1,\dots,\boldsymbol{\theta}_N} \sum_{i=1}^N \frac{n_i}{n} \mathcal{L}_{\hat{\mathcal{D}}_i}(\boldsymbol{\theta}_i)$.

### 4.2 Personalized Representation Bias Memory (PRBM)

Due to the existence of statistical heterogeneity in FL, the local feature extractor intends to learn biased representations after being overwritten by the received global model parameters. To detach and store the representation bias locally, we propose a personalized module PRBM that memorizes representation bias for client $i$. Originally, the feature representation $\boldsymbol{z}_i \in \mathbb{R}^K$ is directly fed into the predictor in Eq. (2). Instead, we consider $\boldsymbol{z}_i$ as the combination of a global $\boldsymbol{z}_i^g \in \mathbb{R}^K$ and a personalized $\bar{\boldsymbol{z}}_i^p \in \mathbb{R}^K$, *i.e.*,

$$\boldsymbol{z}_i := \boldsymbol{z}_i^g + \bar{\boldsymbol{z}}_i^p. \tag{4}$$

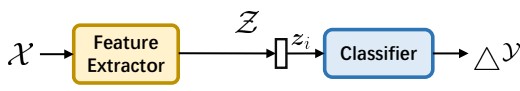

(a) Local model (original).

(b) Local model (ours).

Figure 2: The illustration of the local model. We emphasize the parts that correspond to PRBM and MR with red and green, respectively.

We let the feature extractor output $\boldsymbol{z}_i^g$ instead of the original $\boldsymbol{z}_i$, *i.e.*, $\boldsymbol{z}_i^g := f(\boldsymbol{x}_i; \boldsymbol{\theta}^f)$ and keep the trainable vector $\bar{\boldsymbol{z}}_i^p$ locally. $\bar{\boldsymbol{z}}_i^p$ is specific among clients but identical for all the local data on one client, so it memorizes client-specific mean. The original feature extractor is trained to capture the biased features for $\boldsymbol{z}_i$. Instead, with the personalized mean stored in $\bar{\boldsymbol{z}}_i^p$, the feature extractor turns to capture $\boldsymbol{z}_i^g$ with less biased feature information. We illustrate the difference between the original approach and our method in Figure 2 (PRBM). Then, we define the local objective as $\min_{\boldsymbol{\theta}_i} \mathcal{L}_{\mathcal{D}_i}(\boldsymbol{\theta}_i)$, where $\boldsymbol{\theta}_i := [\boldsymbol{\theta}^f; \bar{\boldsymbol{z}}_i^p; \boldsymbol{\theta}^h]$,

$$\mathcal{L}_{\mathcal{D}_i}(\boldsymbol{\theta}_i) := \mathbb{E}_{(\boldsymbol{x}_i, y_i) \sim \mathcal{D}_i}[\ell(h(f(\boldsymbol{x}_i; \boldsymbol{\theta}^f) + \bar{\boldsymbol{z}}_i^p; \boldsymbol{\theta}^h), y_i)]. \tag{5}$$

From the view of transformation, we rewrite Eq. (5) to

$$\mathcal{L}_{\mathcal{D}_i}(\boldsymbol{\theta}_i) := \mathbb{E}_{(\boldsymbol{x}_i, y_i) \sim \mathcal{D}_i}[\ell(h(\text{PRBM}(f(\boldsymbol{x}_i; \boldsymbol{\theta}^f); \bar{\boldsymbol{z}}_i^p); \boldsymbol{\theta}^h), y_i)], \tag{6}$$

where PRBM $: \mathcal{Z} \mapsto \mathcal{Z}$ a personalized *translation* transformation [78] parameterized by $\bar{\boldsymbol{z}}_i^p$. Formally, $\text{PRBM}(\boldsymbol{z}_i^g; \bar{\boldsymbol{z}}_i^p) = \boldsymbol{z}_i^g + \bar{\boldsymbol{z}}_i^p, \forall \boldsymbol{z}_i^g \in \mathcal{Z}$. With PRBM, we create an additional level of representation $\boldsymbol{z}_i^g$ besides the original level of representation $\boldsymbol{z}_i$. We call $\boldsymbol{z}_i^g$ and $\boldsymbol{z}_i$ as the *first and second levels of representation*, respectively. For the original local model (Figure 2(a)), we have $\boldsymbol{z}_i^g \equiv \boldsymbol{z}_i$.

### 4.3 Mean Regularization (MR)

Without explicit guidance, it is hard for the feature extractor and the *trainable* PRBM to distinguish between unbiased and biased information in representations automatically. Therefore, to let the feature extractor focus on the unbiased information and further separate $\boldsymbol{z}_i^g$ and $\bar{\boldsymbol{z}}_i^p$, we propose an MR that explicitly guides the local feature extractor to generate $\boldsymbol{z}_i^g$ with the help of a client-invariant mean, which is opposite to the client-specific mean memorized in $\bar{\boldsymbol{z}}_i^p$, as shown in Figure 2 (MR). Specifically, we regularize the mean of $\boldsymbol{z}_i^g$ to the consensual global mean $\bar{\boldsymbol{z}}^g$ at each feature dimension independently. We then modify Eq. (6) as

$$\mathcal{L}_{\mathcal{D}_i}(\boldsymbol{\theta}_i) := \mathbb{E}_{(\boldsymbol{x}_i, y_i) \sim \mathcal{D}_i}[\ell(h(\text{PRBM}(f(\boldsymbol{x}_i; \boldsymbol{\theta}^f); \bar{\boldsymbol{z}}_i^p); \boldsymbol{\theta}^h), y_i)] + \kappa \cdot \text{MR}(\bar{\boldsymbol{z}}_i^g, \bar{\boldsymbol{z}}^g), \tag{7}$$

where $\bar{\boldsymbol{z}}_i^g = \mathbb{E}_{(\boldsymbol{x}_i, y_i) \sim \mathcal{D}_i}[f(\boldsymbol{x}_i; \boldsymbol{\theta}^f)]$. We obtain the consensus $\bar{\boldsymbol{z}}^g = \sum_{i=1}^N \bar{\boldsymbol{z}}_i^g$ *during the initialization period before FL (see Algorithm 1)*. We measure the distance of $\bar{\boldsymbol{z}}_i^g$ and $\bar{\boldsymbol{z}}^g$ by mean squared error (MSE) [72], and $\kappa$ is a hyperparameter to control the importance of MR for different tasks. Empirically,

$$\mathcal{L}_{\hat{\mathcal{D}}_i}(\boldsymbol{\theta}_i) := \frac{1}{n_i} \sum_{j=1}^{n_i} \ell(h(\text{PRBM}(f(\boldsymbol{x}_{ij}; \boldsymbol{\theta}^f); \bar{\boldsymbol{z}}_i^p); \boldsymbol{\theta}^h), y_{ij}) + \kappa \cdot \text{MR}(\frac{1}{n_i} \sum_{j=1}^{n_i} f(\boldsymbol{x}_{ij}; \boldsymbol{\theta}^f), \bar{\boldsymbol{z}}^g), \tag{8}$$

which is also optimized by SGD following FedAvg.

In Eq. (8), the value of the MR term is obtained after calculating the empirical version of $\bar{z}_i^g$: $\hat{\bar{z}}_i^g = \frac{1}{n_i} \sum_{j=1}^{n_i} f(\boldsymbol{x}_{ij}; \boldsymbol{\theta}^f)$ over the entire local data, but the loss computing in SGD cannot see all the local data during one forward pass in one batch. In practice, inspired by the widely-used moving average [48, 90] in approximating statistics over data among batches, in each batch, we obtain

$$\hat{\bar{z}}_i^g = (1 - \mu) \cdot \hat{\bar{z}}_{i,old}^g + \mu \cdot \hat{\bar{z}}_{i,new}^g, \tag{9}$$

where $\hat{\bar{z}}_{i,old}^g$ and $\hat{\bar{z}}_{i,new}^g$ are computed in the previous batch and current batch, respectively. $\mu$ is a hyperparameter called momentum that controls the importance of the current batch. The feature extractor is updated continuously during local training but discontinuously between adjacent two iterations due to server aggregation. Thus, we only calculate $\hat{\bar{z}}_i^g$ via Eq. (9) during local training and recalculate it in a new iteration without using its historical records. We consider the representative FedAvg+DBE as an example and show the entire learning process in Algorithm 1.

---

**Algorithm 1** The Learning Process in FedAvg+DBE

---

**Input:** $N$ clients with their local data; initial parameters $\boldsymbol{\theta}^{f,0}$ and $\boldsymbol{\theta}^{h,0}$; $\eta$: local learning rate; $\kappa$ and $\mu$: hyperparameters; $\rho$: client joining ratio; $E$: local epochs; $T$: total communication iterations.
**Output:** Global model parameters $\{\boldsymbol{\theta}^f, \boldsymbol{\theta}^h\}$ and personalized model parameters $\{\bar{z}_1^p, \ldots, \bar{z}_N^p\}$.

▷ *Initialization Period*

1: Server sends $\{\boldsymbol{\theta}^{f,0}, \boldsymbol{\theta}^{h,0}\}$ to all clients to initialize their local models.
2: $N$ clients train their local models *without DBE* for one epoch and collect client-specific mean $\{\bar{z}_1^g, \ldots, \bar{z}_N^g\}$ over their data domain.
3: Server generates a consensual global mean $\bar{z}^g$ through weighted averaging: $\bar{z}^g = \sum_{i=1}^N \frac{n_i}{n} \bar{z}_i^g$.
4: Client $i$ initializes $\bar{z}_i^{p,0}, \forall i \in [N]$.

▷ *Federated Learning Period*

5: **for** communication iteration $t = 1, \ldots, T$ **do**
6:     Server samples a client subset $\mathcal{I}^t$ based on $\rho$.
7:     Server sends $\{\boldsymbol{\theta}^{f,t-1}, \boldsymbol{\theta}^{h,t-1}\}$ to each client in $\mathcal{I}^t$.
8:     **for** Client $i \in \mathcal{I}^t$ in parallel **do**
9:         Initialize $f$ and $h$ with $\boldsymbol{\theta}^{f,t-1}$ and $\boldsymbol{\theta}^{h,t-1}$, respectively.
10:        Obtain $\{\boldsymbol{\theta}_i^{f,t}, \bar{z}_i^{p,t}, \boldsymbol{\theta}_i^{h,t}\}$ using SGD for $\min_{\boldsymbol{\theta}_i} \mathcal{L}_{\hat{\mathcal{D}}_i}(\boldsymbol{\theta}_i)$ with $\eta$, $\kappa$ and $\mu$ for $E$ epochs.
11:        Upload $\{\boldsymbol{\theta}_i^{f,t}, \boldsymbol{\theta}_i^{h,t}\}$ to the server.
12:     Server calculates $n^t = \sum_{i \in \mathcal{I}^t} n_i$ and obtains
13:        $\boldsymbol{\theta}^{f,t} = \sum_{i \in \mathcal{I}^t} \frac{n_i}{n^t} \boldsymbol{\theta}_i^{f,t}$;
14:        $\boldsymbol{\theta}^{h,t} = \sum_{i \in \mathcal{I}^t} \frac{n_i}{n^t} \boldsymbol{\theta}_i^{h,t}$.
15: **return** $\{\boldsymbol{\theta}^{f,T}, \boldsymbol{\theta}^{h,T}\}$ and $\{\bar{z}_1^{p,T}, \ldots, \bar{z}_N^{p,T}\}$

---

### 4.4 Improved Bi-directional Knowledge Transfer

In the FL field, prior methods draw a connection from FL to domain adaptation for theoretical analysis and consider a binary classification problem [21, 55, 69, 96]. The traditional FL methods, which focus on enhancing the performance of a global model, regard local domains $\mathcal{D}_i, i \in [N]$ and the virtual global domain $\mathcal{D}$ as the source domain and the target domain, respectively [96], which is called local-to-global knowledge transfer in this paper. In contrast, pFL methods that focus on improving the performance of personalized models regard $\mathcal{D}$ and $\mathcal{D}_i, i \in [N]$ as the source domain and the target domain, respectively [21, 55, 69]. We call this kind of adaptation as global-to-local knowledge transfer. The local-to-global knowledge transfer happens on the server while the global-to-local one occurs on the client. Please refer to Appendix B for details and proofs.

#### 4.4.1 Local-To-Global Knowledge Transfer

Here, we consider the transfer after the server receives a client model. We guide the feature extractor to learn representations with a global mean and gradually narrow the gap between the local domain and global domain at the first level of representation (*i.e.*, $z_i^g$) to improve knowledge transfer:

**Corollary 1.** *Consider a local data domain $\mathcal{D}_i$ and a virtual global data domain $\mathcal{D}$ for client $i$ and the server, respectively. Let $\mathcal{D}_i = \langle \mathcal{U}_i, c^* \rangle$ and $\mathcal{D} = \langle \mathcal{U}, c^* \rangle$, where $c^* : \mathcal{X} \mapsto \mathcal{Y}$ is a ground-truth*

*labeling function. Let $\mathcal{H}$ be a hypothesis space of VC dimension $d$ and $h : \mathcal{Z} \mapsto \mathcal{Y}, \forall\, h \in \mathcal{H}$. When using* DBE, *given a feature extraction function $\mathcal{F}^g : \mathcal{X} \mapsto \mathcal{Z}$ that shared between $\mathcal{D}_i$ and $\mathcal{D}$, a random labeled sample of size $m$ generated by applying $\mathcal{F}^g$ to a random sample from $\mathcal{U}_i$ labeled according to $c^*$, then for every $h^g \in \mathcal{H}$, with probability at least $1 - \delta$:*

$$\mathcal{L}_{\mathcal{D}}(h^g) \leq \mathcal{L}_{\hat{\mathcal{D}}_i}(h^g) + \sqrt{\frac{4}{m}\left(d \log \frac{2em}{d} + \log \frac{4}{\delta}\right)} + d_{\mathcal{H}}(\tilde{\mathcal{U}}_i^g, \tilde{\mathcal{U}}^g) + \lambda_i,$$

*where $\mathcal{L}_{\hat{\mathcal{D}}_i}$ is the empirical loss on $\mathcal{D}_i$, $e$ is the base of the natural logarithm, and $d_{\mathcal{H}}(\cdot, \cdot)$ is the $\mathcal{H}$-divergence between two distributions. $\lambda_i := \min_{h^g} \mathcal{L}_{\mathcal{D}}(h^g) + \mathcal{L}_{\mathcal{D}_i}(h^g)$, $\tilde{\mathcal{U}}_i^g \subseteq \mathcal{Z}$, $\tilde{\mathcal{U}}^g \subseteq \mathcal{Z}$, and $d_{\mathcal{H}}(\tilde{\mathcal{U}}_i^g, \tilde{\mathcal{U}}^g) \leq d_{\mathcal{H}}(\tilde{\mathcal{U}}_i, \tilde{\mathcal{U}})$. $\tilde{\mathcal{U}}_i^g$ and $\tilde{\mathcal{U}}^g$ are the induced distributions of $\mathcal{U}_i$ and $\mathcal{U}$ under $\mathcal{F}^g$, respectively. $\tilde{\mathcal{U}}_i$ and $\tilde{\mathcal{U}}$ are the induced distributions of $\mathcal{U}_i$ and $\mathcal{U}$ under $\mathcal{F}$, respectively. $\mathcal{F}$ is the feature extraction function in the original FedAvg without* DBE.

As shown in Figure 2, given any $x_i$ on client $i$, one can obtain $z_i$ via $\mathcal{F}$ in original FedAvg or obtain $z_i^g$ via $\mathcal{F}^g$ in FedAvg+DBE. With $d_{\mathcal{H}}(\tilde{\mathcal{U}}_i^g, \tilde{\mathcal{U}}^g) \leq d_{\mathcal{H}}(\tilde{\mathcal{U}}_i, \tilde{\mathcal{U}})$ holds, we can achieve a lower generalization bound in local-to-global knowledge transfer than traditional FL, thus training a better global feature extractor to produce representations with higher quality over all labels. A small gap between the local domain and global domain in $\mathcal{Z}$ promotes the knowledge transfer from clients to the server [82, 92, 94].

### 4.4.2 Global-To-Local Knowledge Transfer

The global-to-local knowledge transfer focuses on the assistance role of the global model parameters for facilitating local training, *i.e.*, the transfer ability from $\mathcal{D}$ to $\mathcal{D}_i$. After the client receives the global model and equips it with PRBM, for the second level of representation (*i.e.*, $z_i$), we have

**Corollary 2.** *Let $\mathcal{D}_i$, $\mathcal{D}$, $\mathcal{F}^g$, and $\lambda_i$ defined as in Corollary 1. Given a translation transformation function* PRBM $: \mathcal{Z} \mapsto \mathcal{Z}$ *that shared between $\mathcal{D}_i$ and virtual $\mathcal{D}$, a random labeled sample of size $m$ generated by applying $\mathcal{F}'$ to a random sample from $\mathcal{U}_i$ labeled according to $c^*$, $\mathcal{F}' =$ PRBM $\circ \mathcal{F}^g :$ $\mathcal{X} \mapsto \mathcal{Z}$, then for every $h' \in \mathcal{H}$, with probability at least $1 - \delta$:*

$$\mathcal{L}_{\mathcal{D}_i}(h') \leq \mathcal{L}_{\hat{\mathcal{D}}}(h') + \sqrt{\frac{4}{m}\left(d \log \frac{2em}{d} + \log \frac{4}{\delta}\right)} + d_{\mathcal{H}}(\tilde{\mathcal{U}}', \tilde{\mathcal{U}}_i') + \lambda_i,$$

*where $d_{\mathcal{H}}(\tilde{\mathcal{U}}', \tilde{\mathcal{U}}_i') = d_{\mathcal{H}}(\tilde{\mathcal{U}}^g, \tilde{\mathcal{U}}_i^g) \leq d_{\mathcal{H}}(\tilde{\mathcal{U}}, \tilde{\mathcal{U}}_i) = d_{\mathcal{H}}(\tilde{\mathcal{U}}_i, \tilde{\mathcal{U}})$. $\tilde{\mathcal{U}}'$ and $\tilde{\mathcal{U}}_i'$ are the induced distributions of $\mathcal{U}$ and $\mathcal{U}_i$ under $\mathcal{F}'$, respectively.*

Given $x_i$ on client $i$, we can obtain $z_i$ via $\mathcal{F}'$ in FedAvg+DBE. $h^g = h' \circ$ PRBM, so PRBM does not influence the value of $d_{\mathcal{H}}(\cdot, \cdot)$ for the pair of $h^g$ and $h'$ (see Appendix B.3), then we have $d_{\mathcal{H}}(\tilde{\mathcal{U}}', \tilde{\mathcal{U}}_i') = d_{\mathcal{H}}(\tilde{\mathcal{U}}^g, \tilde{\mathcal{U}}_i^g)$. The inequality $d_{\mathcal{H}}(\tilde{\mathcal{U}}', \tilde{\mathcal{U}}_i') \leq d_{\mathcal{H}}(\tilde{\mathcal{U}}, \tilde{\mathcal{U}}_i)$ shows that the information aggregated on the server can be more easily transferred to clients with our proposed DBE than FedAvg. We train PRBM on the local loss and preserve it locally, so the local feature extractors can generate representations suitable for clients' personalized tasks. According to Corollary 1 and Corollary 2, adding DBE facilitates the bi-directional knowledge transfer in each iteration, gradually promoting global and local model learning as the number of iterations increases.

### 4.5 Negligible Additional Communication and Computation Overhead

DBE only modifies the local training, so the downloading, uploading, and aggregation processes in FedAvg are unaffected. In FedAvg+DBE, the communication overhead per iteration is the same as FedAvg but requires fewer iterations to converge (see Appendix D). Moreover, PRBM only introduces $K$ additional trainable parameters, and the MSE value in the parameterless MR is computed for two representations of $K$ dimension. $K$ is the representation space dimension, typically a smaller value than the dimension of data inputs or model parameters [8, 83]. Thus, DBE introduces no additional communication overhead and negligible computation overhead for local training in any iteration.

### 4.6 Privacy-Preserving Discussion

Compared to FedAvg, using DBE requires client $i$ to upload one client-specific mean $\bar{z}_i^g$ (one $K$-dimensional vector) to the server ***only once*** before FL, which solely captures the magnitude of the

mean value for each feature dimension within the context of the given datasets and models. Thanks to this particular characteristic, as shown in Section 5.1.4, the performance of FedAvg+DBE can be minimally affected while enhancing its privacy-preserving capabilities by introducing proper Gaussian noise with a zero mean to $\bar{z}_i^g$ during the initialization phase.

# 5  Experiments

**Datasets and models.**  Following prior FL approaches [14, 20, 45, 50, 56], we use four public datasets for classification problems in FL, including three CV datasets: Fashion-MNIST (FM-NIST) [77], Cifar100 [42], and Tiny-ImageNet (100K images with 200 labels) [19], as well as one NLP dataset: AG News [91]. For three CV datasets, we adopt the popular 4-layer CNN by default following FedAvg, which contains two convolution layers (denoted by CONV1 and CONV2) and two fully connected layers (denoted by FC1 and FC2). Besides, we also use a larger model ResNet-18 [29] on Tiny-ImageNet. For AG News, we use the famous text classification model fastText [36].

**Statistically heterogeneous scenarios.**  There are two widely used approaches to construct statistically heterogeneous scenarios on public datasets: the pathological setting [56, 66] and practical setting [45, 50]. For the pathological setting, disjoint data with 2/10/20 labels for each client are sampled from 10/100/200 labels on FMNIST/Cifar100/Tiny-ImageNet with different data amounts. For the practical setting, we sample data from FMNIST, Cifar100, Tiny-ImageNet, and AG News based on the Dirichlet distribution [50] (denoted by $Dir(\beta)$). Specifically, we allocate a $q_{c,i}$ ($q_{c,i} \sim Dir(\beta)$) proportion of samples with label $c$ to client $i$, and we set $\beta = 0.1/\beta = 1$ by default for CV/NLP tasks following previous FL approaches [50, 75].

**Implementation Details.**  Following pFedMe and FedRoD, we have 20 clients and set client participating ratio $\rho = 1$ by default unless otherwise stated. We measure the generic representation quality across clients and evaluate the MDL [65, 74] of representations over all class labels. To simulate the common FL scenario where data only exists on clients, we split the data among each client into two parts: a training set (75% data) and a test set (25% data). Following pFedMe, we evaluate pFL methods by averaging the results of personalized models on the test set of each client and evaluate traditional FL methods by averaging the results of the global model on each client. Following FedAvg, we set the batch size to 10 and the number of local epochs to 1, so the number of local SGD steps is $\lfloor \frac{n_i}{10} \rfloor$ for client $i$. We run three trials for all methods until empirical convergence on each task and report the mean value. For more details and results (*e.g.*, fine-tuning FedAvg on new participants and a real-world application), please refer to Appendix D.

## 5.1  Experimental Study for Adding DBE

### 5.1.1  How to Split the Model?

A model is split into a feature extractor and a classifier, but there are various ways for splitting, as each layer in a deep neural network (DNN) outputs a feature representation and feeds it into the next layer [8, 44] [8, 44]. We focus on inserting DBE between the feature extractor and the classifier, but which splitting way is the best for DBE? Here we answer this question by comparing the results regarding MDL and accuracy when the model is split at each layer in the popular 4-layer CNN. We show the MDL of the representation $z_i^g$ outputted by the prepositive layer of DBE (with underline here) and show MDL of $z_i$ for other layers. Low MDL and high accuracy indicate superior generalization ability and superior personalization ability, respectively.

In Table 1, the generic representation quality is improved at each layer for all splitting ways, which shows that no matter how the model is split, DBE can enhance the generalization ability of the global feature extractor. Among these splitting ways, assigning all FC layers to the classifier, *i.e.*, CONV2→DBE →FC1, achieves almost the lowest MDL and highest accuracy. Meanwhile, FC1→DBE →FC2 can also achieve excellent performance with **only** $4.73\%$ trainable parameters for DBE.

Since FedRep, FedRoD, and FedBABU choose the last FC layer as the classifier by default, we follow them for a fair comparison and insert DBE before the last FC layer (*e.g.*, FC1→DBE →FC2). In Table 1, our FC1→DBE →FC2 outperforms FedPer, FedRep, FedRoD, FedBABU, and FedAvg with lower MDL and higher accuracy. Since feature extractors in FedPer and FedRep are locally trained to cater to personalized classifiers, they extract representations with low quality.

Table 1: The MDL (bits, ↓) of layer-wise representations, test accuracy (%, ↑), and the number of trainable parameters (↓) in PRBM when adding DBE to FedAvg on Tiny-ImageNet using 4-layer CNN in the practical setting. We also show corresponding results for the close pFL methods. For FedBABU, "[36.82]" indicates the test accuracy after post-FL fine-tuning for 10 local epochs.

| Metrics | MDL | | | | Accuracy | Param. |
|---|---|---|---|---|---|---|
| | CONV1→CONV2 | CONV2→FC1 | FC1→FC2 | Logits | | |
| FedPer [3] | 5143 | 4574 | 3885 | 4169 | 33.84 | — |
| FedRep [20] | 5102 | 4237 | 3922 | 4244 | 37.27 | — |
| FedRoD [14] | 5063 | 4264 | 3783 | 3820 | 36.43 | — |
| FedBABU [61] | 5083 | 4181 | 3948 | 3849 | 16.86 [36.82] | — |
| Original (FedAvg) | 5081 | 4151 | 3844 | 3895 | 19.46 | 0 |
| CONV1→DBE →CONV2 | 4650 (-8.48%) | 4105 (-1.11%) | 3679 (-4.29%) | 3756 (-3.57%) | 21.81 (+2.35) | 28800 |
| CONV2→DBE →FC1 | **4348 (-14.43%)** | 3716 (-10.48%) | **3463 (-9.91%)** | **3602 (-7.52%)** | 47.03 (+27.57) | 10816 |
| FC1→DBE →FC2 | 4608 (-9.31%) | **3689 (-11.13%)** | 3625 (-5.70%) | 3688 (-5.31%) | 43.32 (+23.86) | 512 |

### 5.1.2 Representation Bias Eliminated for the First Level of Representation

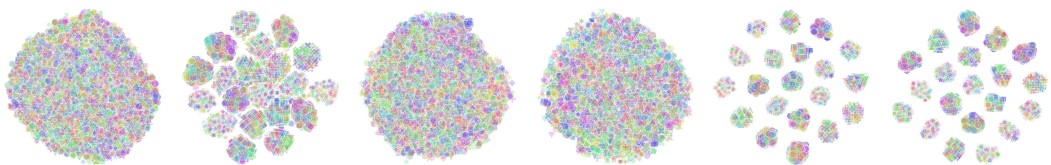

(a) FedAvg (B). (b) FedAvg (A). (c) +DBE ($z_i^g$, B). (d) +DBE ($z_i^g$, A). (e) +DBE ($z_i$, B). (f) +DBE ($z_i$, A).

Figure 3: t-SNE visualization for representations on Tiny-ImageNet (200 labels). "B" and "A" denote "before local training" and "after local training", respectively. We use *color* and *shape* to distinguish *labels* and *clients*, respectively. *Best viewed in color and zoom-in.*

We visualize the feature representations using t-SNE [73] in Figure 3. Compared to the representations outputted by the feature extractor in FedAvg, $z_i^g$ in FedAvg+DBE is no longer biased to the local data domain of each client after local training. With the personalized translation transformation PRBM, $z_i$ can fit the local domain of each client either before or after local training. According to Figure 3(b), Figure 3(e) and Figure 3(f), $z_i$ in FedAvg+DBE can fit the local domain better than FedAvg.

### 5.1.3 Ablation Study for DBE

Table 2: The MDL (bits, ↓) and test accuracy (%, ↑) when adding DBE to FedAvg on Tiny-ImageNet using 4-layer CNN and ResNet-18 in the practical setting.

| Models | 4-layer CNN | | | | ResNet-18 | | | |
|---|---|---|---|---|---|---|---|---|
| **Components** | FedAvg | +MR | +PRBM | +DBE | FedAvg | +MR | +PRBM | +DBE |
| MDL | 3844 | 3643 | 3699 | **3625** | 3560 | 3460 | 3471 | **3454** |
| Accuracy | 19.46 | 22.21 | 26.70 | **43.32** | 19.45 | 20.85 | 38.27 | **42.98** |

We further study the contribution of MR and PRBM in terms of generalization and personalization abilities by applying only one of them to FedAvg. From Table 2, we find that for 4-layer CNN and ResNet-18, +DBE gives a larger improvement in both MDL and accuracy than just using MR or PRBM, which suggests that MR and PRBM can boost each other in bi-directional knowledge transfer. The contribution of MR is greater than that of PRBM in improving the generalization ability in MDL, while +PRBM gains more accuracy improvement for personalization ability than MR.

### 5.1.4 Privacy-Preserving Ability

Following FedPAC [70], we add Gaussian noise to client-specific means $\bar{z}_1^g, \ldots, \bar{z}_N^g$ with a scale parameter ($s$) for the noise distribution and perturbation coefficient ($q$) for the noise. Adding the unbiased noise sampled from one distribution is beneficial for representation bias elimination and can further improve the performance of DBE to some extent, as shown in Table 3. Besides, adding too much noise can also bring an accuracy decrease. However, setting $s = 0.05$ and $q = 0.2$ is sufficient to ensure privacy protection according to FedPCL.

Table 3: The test accuracy (%, ↑) using FedAvg+DBE on TINY in the practical setting with noise.

| | $q = 0.2$ | | | | $s = 0.05$ | | | |
| Original | $s = 0.05$ | $s = 0.5$ | $s = 1$ | $s = 5$ | $q = 0.1$ | $q = 0.5$ | $q = 0.8$ | $q = 0.9$ |
|---|---|---|---|---|---|---|---|---|
| 43.32 | 44.10 | 44.15 | 43.78 | 36.27 | 43.81 | **44.45** | 43.30 | 41.75 |

### 5.1.5 DBE Improves Other Traditional Federated Learning Methods

Table 4: The MDL (bits, ↓) and test accuracy (%, ↑) before and after adding DBE to traditional FL methods on Cifar100, Tiny-ImageNet, and AG News in the practical setting. TINY and TINY* represent using 4-layer CNN and ResNet-18 on Tiny-ImageNet, respectively.

| Metrics | MDL | | | | Accuracy | | | |
|---|---|---|---|---|---|---|---|---|
| **Datasets** | Cifar100 | TINY | TINY* | AG News | Cifar100 | TINY | TINY* | AG News |
| SCAFFOLD [38] | 1499 | 3661 | 3394 | 1931 | 33.08 | 23.26 | 24.90 | 88.13 |
| FedProx [46] | 1523 | 3701 | 3570 | 2092 | 31.99 | 19.37 | 19.27 | 87.21 |
| MOON [45] | 1516 | 3696 | 3536 | 1836 | 32.37 | 19.68 | 19.02 | 84.14 |
| FedGen [96] | 1506 | 3675 | 3551 | 1414 | 30.96 | 19.39 | 18.53 | 89.86 |
| SCAFFOLD+DBE | **1434** | **3549** | **3370** | **1743** | **63.61** | **45.55** | **45.09** | **96.73** |
| FedProx+DBE | **1439** | **3587** | **3490** | **1689** | **63.22** | **42.28** | **41.45** | **96.62** |
| MOON+DBE | **1432** | **3580** | **3461** | **1683** | **63.26** | **43.43** | **41.10** | **96.68** |
| FedGen+DBE | **1426** | **3563** | **3488** | **1098** | **63.26** | **42.54** | **41.87** | **97.16** |

A large number of FL methods design algorithms based on the famous FedAvg [37, 56, 69]. Although we describe DBE based on FedAvg for example, DBE can also be applied to other traditional FL methods to improve their generalization and personalization abilities. Here, we apply DBE to another four representative traditional FL methods: SCAFFOLD [38], FedProx [46], MOON [45], and FedGen [96]. They belong to four categories: update-correction-based FL, regularization-based FL, model-split-based FL, and knowledge-distillation-based FL, respectively. In Table 4, DBE promotes traditional FL methods by at most **-22.35%** in MDL (bits) and **+32.30** in accuracy (%), respectively. Based on the results of Table 2 and Table 4 on Tiny-ImageNet, FedAvg+DBE achieves lower MDL and higher accuracy than close methods MOON and FedGen.

## 5.2 Comparison with Personalized Federated Learning Methods

### 5.2.1 Personalization Ability on Various Datasets

Table 5: The test accuracy (%, ↑) of pFL methods in two statistically heterogeneous settings. Cifar100$^\dagger$ represents the experiment with 100 clients and joining ratio $\rho = 0.5$ on Cifar100.

| Settings | Pathological setting | | | Practical setting | | | | | |
|---|---|---|---|---|---|---|---|---|---|
| | FMNIST | Cifar100 | TINY | FMNIST | Cifar100 | Cifar100$^\dagger$ | TINY | TINY* | AG News |
| Per-FedAvg [22] | 99.18 | 56.80 | 28.06 | 95.10 | 44.28 | 38.28 | 25.07 | 21.81 | 87.08 |
| pFedMe [67] | 99.35 | 58.20 | 27.71 | 97.25 | 47.34 | 31.13 | 26.93 | 33.44 | 87.08 |
| Ditto [47] | 99.44 | 67.23 | 39.90 | 97.47 | 52.87 | 39.01 | 32.15 | 35.92 | 91.89 |
| FedPer [3] | 99.47 | 63.53 | 39.80 | 97.44 | 49.63 | 41.21 | 33.84 | 38.45 | 91.85 |
| FedRep [20] | 99.56 | 67.56 | 40.85 | 97.56 | 52.39 | 41.51 | 37.27 | 39.95 | 92.25 |
| FedRoD [14] | 99.52 | 62.30 | 37.95 | 97.52 | 50.94 | 48.56 | 36.43 | 37.99 | 92.16 |
| FedBABU [61] | 99.41 | 66.85 | 40.72 | 97.46 | 55.02 | 52.07 | 36.82 | 34.50 | 95.86 |
| APFL [21] | 99.41 | 64.26 | 36.47 | 97.25 | 46.74 | 39.47 | 34.86 | 35.81 | 89.37 |
| FedFomo [89] | 99.46 | 62.49 | 36.55 | 97.21 | 45.39 | 37.59 | 26.33 | 26.84 | 91.20 |
| APPLE [52] | 99.30 | 65.80 | 36.22 | 97.06 | 53.22 | — | 35.04 | 39.93 | 84.10 |
| FedAvg | 80.41 | 25.98 | 14.20 | 85.85 | 31.89 | 28.81 | 19.46 | 19.45 | 87.12 |
| FedAvg+DBE | **99.74** | **73.38** | **42.89** | **97.69** | **64.39** | **63.43** | **43.32** | **42.98** | **96.87** |

To further show the superiority of the DBE-equipped traditional FL methods to existing pFL methods, we compare the representative FedAvg+DBE with ten SOTA pFL methods, as shown in Table 5. Note that APPLE is designed for cross-silo scenarios and assumes $\rho = 1$. For Per-FedAvg and FedBABU, we show the test accuracy after post-FL fine-tuning. FedAvg+DBE improves FedAvg at most **+47.40**

on Cifar100 in the pathological setting and outperforms the best SOTA pFL methods by up to **+11.36** on Cifar100[†] including the fine-tuning-based methods that require additional post-FL effort.

### 5.2.2 Personalization Ability Under Various Heterogeneous Degrees

Following prior methods [45, 50], we also evaluate FedAvg+DBE with different $\beta$ on Tiny-ImageNet using 4-layer CNN to study the influence of heterogeneity, as shown in Table 6. Most pFL methods are specifically designed for extremely heterogeneous scenarios and can achieve high accuracy at $\beta = 0.01$, but some of them cannot maintain the advantage compared to FedAvg in moderate scenarios. However, FedAvg+DBE can automatically adapt to all these scenarios without tuning.

Table 6: The test accuracy (%, ↑) and computation overhead (↓) of pFL methods.

| Items | Heterogeneity | | | pFL+MR | | Overhead | |
|---|---|---|---|---|---|---|---|
| | $\beta = 0.01$ | $\beta = 0.5$ | $\beta = 5$ | Accuracy | Improvement | Total time | Time/iteration |
| Per-FedAvg [22] | 39.39 | 21.14 | 12.08 | — | — | 121 min | 3.56 min |
| pFedMe [67] | 41.45 | 17.48 | 4.03 | — | — | 1157 min | 10.24 min |
| Ditto [47] | 50.62 | 18.98 | 21.79 | 42.82 | 10.67 | 318 min | 11.78 min |
| FedPer [3] | 51.83 | 17.31 | 9.61 | 41.78 | 7.94 | 83 min | 1.92 min |
| FedRep [20] | 55.43 | 16.74 | 8.04 | 41.28 | 4.01 | 471 min | 4.09 min |
| FedRoD [14] | 49.17 | 23.23 | 16.71 | 42.74 | 6.31 | 87 min | 1.74 min |
| FedBABU [61] | 53.97 | 23.08 | 15.42 | 38.17 | 1.35 | 811 min | 1.58 min |
| APFL [21] | 49.96 | 23.31 | 16.12 | 39.22 | 4.36 | 156 min | 2.74 min |
| FedFomo [89] | 46.36 | 11.59 | 14.86 | 29.51 | 3.18 | 193 min | 2.72 min |
| APPLE [52] | 47.89 | 24.24 | 17.79 | — | — | 132 min | 2.93 min |
| FedAvg | 15.70 | 21.14 | 21.71 | — | — | 365 min | 1.59 min |
| FedAvg+DBE | **57.52** | **32.61** | **25.55** | — | — | 171 min | 1.60 min |

### 5.2.3 MR Improves Personalized Federated Learning Methods

Since pFL methods already create personalized models or modules in their specific ways, applying personalized PRBM to the local model might be against their philosophy. To prevent this, we only apply the MR to pFL methods. Besides, the local training schemes (*e.g.*, meta-learning) in Per-FedAvg, pFedMe, and APPLE are different from the simple SGD in FedAvg, which requires modification of the mean calculation in MR, so we do not apply MR to them. According to Corollary 1, MR can promote the local-to-global knowledge transfer between server and client. Therefore, pFL methods can benefit more from a better global model achieving higher accuracy on Tiny-ImageNet with the 4-layer CNN, as shown in Table 6. However, their MR-equipped variants perform worse than FedAvg+DBE (Table 5, TINY) since the representation bias still exists without using PRBM.

### 5.2.4 Computation Overhead

We evaluate FedAvg+DBE in total time and time per iteration on Tiny-ImageNet using ResNet-18, as shown in Table 6. The evaluation task for one method monopolizes one identical machine. FedAvg, FedBABU, and FedAvg+DBE cost almost the same and have the lowest time per iteration among these methods, but FedAvg+DBE requires less total time than FedAvg and FedBABU. Note that the fine-tuning time for FedBABU is not included in Table 6. Since pFedMe and Ditto train an additional personalized model on each client, they cost plenty of time per iteration.

## 6 Conclusion

Due to the naturally existing statistical heterogeneity and the biased local data domains on each client, FL suffers from representation bias and representation degeneration problems. To improve the generalization and personalization abilities for FL, we propose a general framework DBE including two modules PRBM and MR, with a theoretical guarantee. Our DBE can promote the bi-directional knowledge transfer in each iteration, thus improving both generalization and personalization abilities. Besides, we conduct extensive experiments to show the general applicability of DBE to existing FL methods and the superiority of the representative FedAvg+DBE to ten SOTA pFL methods in various scenarios.

## Acknowledgments and Disclosure of Funding

This work was partially supported by the Program of Technology Innovation of the Science and Technology Commission of Shanghai Municipality (Granted No. 21511104700 and 22DZ1100103). This work was also supported in part by the Shanghai Key Laboratory of Scalable Computing and Systems, National Key R&D Program of China (2022YFB4402102), Internet of Things special subject program, China Institute of IoT (Wuxi), Wuxi IoT Innovation Promotion Center (2022SP-T13-C), Industry-university-research Cooperation Funding Project from the Eighth Research Institute in China Aerospace Science and Technology Corporation (Shanghai) (USCAST2022-17), Intel Corporation (UFunding 12679), and the cooperation project from Ant Group ("Wasm-enabled Managed language in security restricted scenarios").

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

We provide more details and results about our work in the appendices. Here are the contents:

- Appendix A: The extended version of the Related Work section in the main body.
- Appendix B: Proofs of Corollary 1 and Corollary 2.
- Appendix C: More details about experimental settings.
- Appendix D: Additional experiments (*e.g.*, a real-world application).
- Appendix E: Broader impacts of our proposed method.
- Appendix F: Limitations of our proposed method.
- Appendix G: Data distribution visualizations for different scenarios in our experiments.

# A   Related Work

As the number of users and sensors rapidly increases with massive growing services on the Internet, the privacy concerns about private data also draw increasing attention of researchers [37, 68, 71]. Then a new distributed machine learning paradigm, federated learning (FL), comes along with the privacy-preserving and collaborative learning abilities [37, 56, 79]. Although there are horizontal FL [46, 56, 79], vertical FL [53, 63, 79], federated transfer learning[15, 51], *etc.*, we focus on the popular horizontal FL and call it FL for short in this paper.

Traditional FL methods concentrate on learning a single global model among a server and clients, but it suffers an accuracy decrease under statistically heterogeneous scenarios, which are common scenarios in practice [47, 56, 67, 86]. Then, many FL methods propose learning personalized models (or modules) for each client besides learning the global model. These FL methods are specifically called personalized FL (pFL) methods [20, 22, 69].

## A.1   Traditional Federated Learning

FL methods perform machine learning through iterative communication and computation on the server and clients. To begin with, we describe the FL procedure in one iteration based on FedAvg [56], which is a famous FL method and a basic framework for later FL methods. The FL procedure includes five stages: (1) A server selects a group of clients to join FL in this iteration and sends the current global model to them; (2) these clients receive the global model and initialize their local model by overwriting their local model with the parameters in the global model; (3) these clients train their local models on their own private local data, respectively; (4) these clients send the trained local models to the server; (5) the server receives client models and aggregates them through weighted averaging on model parameters to obtain a new global model.

Then, massive traditional FL methods are proposed in the literature to improve FedAvg regarding privacy-preserving [49, 58, 95], accuracy [38, 45, 96], fairness [32, 80], overhead [28, 41, 54], *etc*. Here, we focus on the representative traditional FL methods that handle the heterogeneity issues in four categories: update-correction-based FL [25, 38, 60], regularization-based FL [1, 17, 40, 46], model-split-based FL [35, 45], and knowledge-distillation-based FL [27, 33, 88, 96].

Among **update-correction-based FL** methods, SCAFFOLD [38] witnesses the client-drift phenomenon of FedAvg under statistically heterogeneous scenarios due to local training and proposes correcting local update through control variates for each model parameter. Among **regularization-based FL** methods, FedProx [46] modifies the local objective on each client by adding a regularization term to keep local model parameters close to the global model during local training in an element-wise manner. Among **model-split-based FL** methods, MOON [45] observes that local training degenerates representation quality, so it adds a contrastive learning term to let the representations outputted by the local feature extractor be close to the ones outputted by the received global feature extractor given each input during local training. However, input-wise contrastive learning relies on biased local data domains, so MOON still suffers from representation bias. Among **knowledge-distillation-based FL** methods, FedGen [96] learns a generator on the server to produce additional representations, shares the generator among clients, and locally trains the classifier with the combination of the representations outputted by the local feature extractor and the additionally generated representations. In this way, FedGen can reduce the heterogeneity among clients with the augmented representations from the shared generator via knowledge distillation. However, it only considers the local-to-global knowledge transfer for the single global model learning and additionally brings communication and computation overhead for learning and transmitting the generator.

## A.2   Personalized Federated Learning

Different from traditional FL, pFL additionally learns personalized models (or modules) besides the global model. In this paper, we consider pFL methods in four categories: meta-learning-based pFL [13, 22], regularization-based pFL [47, 67], personalized-aggregation-based pFL [21, 52, 89], and model-split-based pFL [3, 14, 20, 61].

**Meta-learning-based pFL.** Meta-learning is a technique that trains deep neural networks (DNNs) on a given dataset for quickly adapting to other datasets with only a few steps of fine-tuning, *e.g.*, MAML [24]. By integrating MAML into FL, Per-FedAvg [22] updates the local models like MAML to capture the learning trends of each client and then aggregates the learning trends by averaging on the server. It obtains personalized models by fine-tuning the global model for each client. Similar to Per-FedAvg, FedMeta [13] also introduces MAML on each client during training and fine-tuning the global model for evaluation. However, it is hard for these meta-learning-based pFL methods to find a consensus learning trend through averaging under statistically heterogeneous scenarios.

**Regularization-based pFL.** Like FedProx, pFedMe [67] and Ditto [47] also utilize the regularization technique, but they modify the objective for additional personalized model training rather than the one for local model training. In pFedMe and Ditto, each client owns two models: the local model that is trained for global model aggregation and the personalized model that is trained for personalization. Specifically, pFedMe regularizes the model parameters between the personalized model and the local model during training while Ditto regularizes the model parameters between the personalized model and the received global model. Besides, Ditto simply trains the local model similar to FedAvg while pFedMe trains the local model based on the personalized model. Although the local model is initialized by the global model, but the initialized local model gradually loses global information during local training. Thus, the personalized model in Ditto can be aware of more global information than the one in pFedMe. Both pFedMe and Ditto require additional memory space to store the personalized model and double the computation resources at least to train both the local model and the personalized model.

**Personalized-aggregation-based pFL.** These pFL methods adaptively aggregate the global model and local model according to the local data on each client, *e.g.*, APFL [21], or directly generate the personalized model using other client models through personalized aggregation on each client, *e.g.*, FedFomo [89] and APPLE [52]. Specifically, APFL aggregates the parameters in the global model and the local model with weighted averaging and adaptively updates the scalar weight based on the gradients. On each client, FedFomo generates the client-specific aggregating weights for the received client models through first-order approximation while APPLE adaptively learns these weights based on the local data. Both FedFomo and APPLE require multiple communication overhead than other FL methods, but FedFomo costs less computation overhead than APPLE attributed to approximation.

**Model-split-based pFL.** These pFL methods split a given model into a feature extractor and a classifier. They treat the feature extractor and the classifier differently. Concretely, FedPer [3] and FedRep [20] keep the classifier locally on each client. FedPer trains the feature extractor and the classifier together while FedRep first fine-tunes the classifier and then trains the feature extractor in each iteration. For FedPer and FedRep, the feature extractor intends to extract representations to cater to these personalized classifiers, thus reducing the generic representation quality. FedRoD [14] trains the local model with the balanced softmax (BSM) loss function [64] and simultaneously learns an additional personalized classifier for each client. However, the BSM loss is useless for missing labels on each client while label missing is a common situation in statistically heterogeneous scenarios [50, 86, 89]. Moreover, the uniform label distribution modified by the BSM cannot reflect the original distribution. The above pFL methods learn personalized models (or modules) in FL, but FedBABU [61] firstly trains the global feature extractor with the frozen classifier during the FL process, then it fine-tunes the global model on each client after FL to obtain personalized models. However, this post-FL fine-tuning is beyond the scope of FL. Almost all the FL methods have multiple fine-tuning variants, *e.g.*, fine-tuning the whole model or only a part of the model. Furthermore, training the feature extractor with the naive and randomly initialized classifier in FL has an uncontrollable risk due to randomness.

# B  Theoretical Derivations

## B.1  Notations and Preliminaries

Following prior arts [21, 55, 69, 96], we consider a binary classification problem in FL here. Recall that $\mathcal{X} \subset \mathbb{R}^D$ is an input space, $\mathcal{Z} \subset \mathbb{R}^K$ is a representation space, and $\mathcal{Y} \subset \{0, 1\}$ is a label space. Let $\mathcal{F} : \mathcal{X} \mapsto \mathcal{Z}$ be a representation function that maps from the input space to the representation space. We denote $\mathcal{D} := \langle \mathcal{U}, c^* \rangle$ as a data domain where the distribution $\mathcal{U} \subseteq \mathcal{X}$ and $c^* : \mathcal{X} \mapsto \mathcal{Y}$ is a ground-truth labeling function. $\tilde{\mathcal{U}}$ is the induced distribution of $\mathcal{U}$ over the representation space $\mathcal{Z}$ under $\mathcal{F}$ [6], *i.e.*, $\tilde{\mathcal{U}} \subseteq \mathcal{Z}$, that satisfies

$$\mathbb{E}_{\boldsymbol{z} \sim \tilde{\mathcal{U}}} \left[ \mathcal{B} \left( \boldsymbol{z} \right) \right] = \mathbb{E}_{\boldsymbol{x} \sim \mathcal{U}} \left[ \mathcal{B} \left( \mathcal{F} \left( \boldsymbol{x} \right) \right) \right], \tag{10}$$

where $\mathcal{B}$ is a probability event. Given fixed but unknown $\mathcal{U}$ and $c^*$, the learning task on one domain is to choose a representation function $\mathcal{F}$ and a hypothesis class $\mathcal{H} \subseteq \{h : \mathcal{Z} \mapsto \mathcal{Y}\}$ to approximate the function $c^*$.

Then, we provide the definition and theorem from Ben-David et al. [6, 7], Blitzer et al. [9], Kifer et al. [39] under their assumptions:

**Definition 1.** *If a space $\mathcal{Z}$ with $\tilde{\mathcal{U}}^a$ and $\tilde{\mathcal{U}}^b$ distributions over $\mathcal{Z}$, let $\mathcal{H}$ be a hypothesis class on $\mathcal{Z}$ and $\mathcal{Z}_h \subseteq \mathcal{Z}$ be the subset with characteristic function $h$, the $\mathcal{H}$-divergence between $\tilde{\mathcal{U}}^a$ and $\tilde{\mathcal{U}}^b$ is*

$$d_{\mathcal{H}}\left(\tilde{\mathcal{U}}^a, \tilde{\mathcal{U}}^b\right) = 2 \sup_{h \in \mathcal{H}} \left| \mathrm{Pr}_{\tilde{\mathcal{U}}^a}\left[\mathcal{Z}_h\right] - \mathrm{Pr}_{\tilde{\mathcal{U}}^b}\left[\mathcal{Z}_h\right] \right|,$$

*where $\mathcal{Z}_h = \{\boldsymbol{z} \in \mathcal{Z} : h(\boldsymbol{z}) = 1\}, h \in \mathcal{H}$.*

Definition 1 implies that $d_{\mathcal{H}}\left(\tilde{\mathcal{U}}^a, \tilde{\mathcal{U}}^b\right) = d_{\mathcal{H}}\left(\tilde{\mathcal{U}}^b, \tilde{\mathcal{U}}^a\right)$.

**Theorem 1.** *Consider a source domain $\mathcal{D}_S$ and a target domain $\mathcal{D}_T$. Let $\mathcal{D}_S = \langle \mathcal{U}_S, c^* \rangle$ and $\mathcal{D}_T = \langle \mathcal{U}_T, c^* \rangle$, where $\mathcal{U}_S \subseteq \mathcal{X}, \mathcal{U}_T \subseteq \mathcal{X}$, and $c^* : \mathcal{X} \mapsto \mathcal{Y}$ is a ground-truth labeling function. Let $\mathcal{H}$ be a hypothesis space of VC dimension $d$ and $h : \mathcal{Z} \mapsto \mathcal{Y}, \forall h \in \mathcal{H}$. Given a feature extraction function $\mathcal{F} : \mathcal{X} \mapsto \mathcal{Z}$ that shared between $\mathcal{D}_S$ and $\mathcal{D}_T$, a random labeled sample of size $m$ generated by applying $\mathcal{F}$ to a random sample from $\mathcal{U}_S$ labeled according to $c^*$, then for every $h \in \mathcal{H}$, with probability at least $1 - \delta$:*

$$\mathcal{L}_{\mathcal{D}_T}(h) \leq \mathcal{L}_{\hat{\mathcal{D}}_S}(h) + \sqrt{\frac{4}{m}\left(d \log \frac{2em}{d} + \log \frac{4}{\delta}\right)} + d_{\mathcal{H}}\left(\tilde{\mathcal{U}}_S, \tilde{\mathcal{U}}_T\right) + \lambda,$$

*where $\mathcal{L}_{\hat{\mathcal{D}}_S}$ is the empirical loss on $\mathcal{D}_S$, $e$ is the base of the natural logarithm, and $d_{\mathcal{H}}(\cdot, \cdot)$ is the $\mathcal{H}$-divergence between two distributions. $\tilde{\mathcal{U}}_S$ and $\tilde{\mathcal{U}}_T$ are the induced distributions of $\mathcal{U}_S$ and $\mathcal{U}_T$ under $\mathcal{F}$, respectively, s.t. $\mathbb{E}_{\boldsymbol{z} \sim \tilde{\mathcal{U}}_S}[\mathcal{B}(\boldsymbol{z})] = \mathbb{E}_{\boldsymbol{x} \sim \mathcal{U}_S}[\mathcal{B}(\mathcal{F}(\boldsymbol{x}))]$ given a probability event $\mathcal{B}$, and so for $\tilde{\mathcal{U}}_T$. $\tilde{\mathcal{U}}_S \subseteq \mathcal{Z}$ and $\tilde{\mathcal{U}}_T \subseteq \mathcal{Z}$. $\lambda := \min_h \mathcal{L}_{\mathcal{D}_S}(h) + \mathcal{L}_{\mathcal{D}_T}(h)$ denotes an oracle performance.*

The traditional FL methods, which focus on enhancing the performance of a global model, regard local domains $\mathcal{D}_i, i \in [N]$ and the virtual global domain $\mathcal{D}$ as the source domain and the target domain, respectively [96], which is called local-to-global knowledge transfer in this paper. In contrast, pFL methods that focus on improving the performance of personalized models regard $\mathcal{D}$ and $\mathcal{D}_i, i \in [N]$ as the source domain and the target domain, respectively [21, 55, 69]. We call this kind of adaptation global-to-local knowledge transfer. The local-to-global knowledge transfer happens on the server while the global-to-local one occurs on the client.

## B.2 Derivations of Corollary 1

As we focus on the local-to-global knowledge transfer on the *server side*, in the FL scenario, we can rewrite Theorem 1 to

**Theorem 2.** *Consider a local data domain $\mathcal{D}_i$ and a virtual global data domain $\mathcal{D}$. Let $\mathcal{D}_i = \langle \mathcal{U}_i, c^* \rangle$ and $\mathcal{D} = \langle \mathcal{U}, c^* \rangle$, where $\mathcal{U}_i \subseteq \mathcal{X}$ and $\mathcal{U} \subseteq \mathcal{X}$. Given a feature extraction function $\mathcal{F} : \mathcal{X} \mapsto \mathcal{Z}$ that shared between $\mathcal{D}_i$ and $\mathcal{D}$, a random labeled sample of size $m$ generated by applying $\mathcal{F}$ to a random sample from $\mathcal{U}_i$ labeled according to $c^*$, then for every $h \in \mathcal{H}$, with probability at least $1 - \delta$:*

$$\mathcal{L}_{\mathcal{D}}(h) \leq \mathcal{L}_{\hat{\mathcal{D}}_i}(h) + \sqrt{\frac{4}{m}\left(d \log \frac{2em}{d} + \log \frac{4}{\delta}\right)} + d_{\mathcal{H}}\left(\tilde{\mathcal{U}}_i, \tilde{\mathcal{U}}\right) + \lambda_i,$$

*where $\tilde{\mathcal{U}}_i$ and $\tilde{\mathcal{U}}$ are the induced distributions of $\mathcal{U}_i$ and $\mathcal{U}$ under $\mathcal{F}$, respectively. $\tilde{\mathcal{U}}_i \subseteq \mathcal{Z}$ and $\tilde{\mathcal{U}} \subseteq \mathcal{Z}$. $\lambda_i := \min_h \mathcal{L}_{\mathcal{D}_i}(h) + \mathcal{L}_{\mathcal{D}}(h)$ denotes an oracle performance.*

**Corollary 1.** *Consider a local data domain $\mathcal{D}_i$ and a virtual global data domain $\mathcal{D}$ for client $i$ and the server, respectively. Let $\mathcal{D}_i = \langle \mathcal{U}_i, c^* \rangle$ and $\mathcal{D} = \langle \mathcal{U}, c^* \rangle$, where $c^* : \mathcal{X} \mapsto \mathcal{Y}$ is a ground-truth labeling function. Let $\mathcal{H}$ be a hypothesis space of VC dimension $d$ and $h : \mathcal{Z} \mapsto \mathcal{Y}, \forall h \in \mathcal{H}$. When using DBE, given a feature extraction function $\mathcal{F}^g : \mathcal{X} \mapsto \mathcal{Z}$ that shared between $\mathcal{D}_i$ and $\mathcal{D}$, a random labeled sample of size $m$ generated by applying $\mathcal{F}^g$ to a random sample from $\mathcal{U}_i$ labeled according to $c^*$, then for every $h^g \in \mathcal{H}$, with probability at least $1 - \delta$:*

$$\mathcal{L}_{\mathcal{D}}(h^g) \leq \mathcal{L}_{\hat{\mathcal{D}}_i}(h^g) + \sqrt{\frac{4}{m}\left(d \log \frac{2em}{d} + \log \frac{4}{\delta}\right)} + d_{\mathcal{H}}\left(\tilde{\mathcal{U}}_i^g, \tilde{\mathcal{U}}^g\right) + \lambda_i,$$

*where $\mathcal{L}_{\hat{\mathcal{D}}_i}$ is the empirical loss on $\mathcal{D}_i$, $e$ is the base of the natural logarithm, and $d_{\mathcal{H}}(\cdot, \cdot)$ is the $\mathcal{H}$-divergence between two distributions. $\lambda_i := \min_{h^g} \mathcal{L}_{\mathcal{D}}(h^g) + \mathcal{L}_{\mathcal{D}_i}(h^g)$, $\tilde{\mathcal{U}}_i^g \subseteq \mathcal{Z}$, $\tilde{\mathcal{U}}^g \subseteq \mathcal{Z}$, and $d_{\mathcal{H}}\left(\tilde{\mathcal{U}}_i^g, \tilde{\mathcal{U}}^g\right) \leq d_{\mathcal{H}}\left(\tilde{\mathcal{U}}_i, \tilde{\mathcal{U}}\right)$. $\tilde{\mathcal{U}}_i^g$ and $\tilde{\mathcal{U}}^g$ are the induced distributions of $\mathcal{U}_i$ and $\mathcal{U}$ under $\mathcal{F}^g$, respectively. $\tilde{\mathcal{U}}_i$ and $\tilde{\mathcal{U}}$ are the induced distributions of $\mathcal{U}_i$ and $\mathcal{U}$ under $\mathcal{F}$, respectively. $\mathcal{F}$ is the feature extraction function in the original FedAvg without DBE.*

*Proof.* Computing $d_{\mathcal{H}}(\cdot, \cdot)$ is identical to learning a classifier to achieve a minimum error of discriminating between points sampled from $\tilde{\mathcal{U}}$ and $\tilde{\mathcal{U}}'$, *i.e.*, a binary domain classification problem [6, 7]. The more difficult

the domain classification problem is, the smaller $d_{\mathcal{H}}\left(\cdot,\cdot\right)$ is. Unfortunately, computing the error of the optimal hyperplane classifier for arbitrary distributions is a well-known NP-hard problem [5, 6]. Thus, researchers approximate the error by learning a linear classifier for the binary domain classification [5, 9, 10]. Inspired by previous approaches [4, 43, 57], we consider using Linear Discriminant Analysis (LDA) for the binary domain classification. The discrimination ability of LDA is measured by the Fisher discriminant ratio (F1) [11, 30, 76]

$$F1\left(\tilde{\mathcal{U}}^a,\tilde{\mathcal{U}}^b\right) = \max_k \left[ \frac{\left(\boldsymbol{\mu}_{\tilde{\mathcal{U}}^a}^k - \boldsymbol{\mu}_{\tilde{\mathcal{U}}^b}^k\right)^2}{\left(\boldsymbol{\sigma}_{\tilde{\mathcal{U}}^a}^k\right)^2 + \left(\boldsymbol{\sigma}_{\tilde{\mathcal{U}}^b}^k\right)^2} \right],$$

where $\boldsymbol{\mu}_{\tilde{\mathcal{U}}^a}^k$ and $\left(\boldsymbol{\sigma}_{\tilde{\mathcal{U}}^a}^k\right)^2$ are the mean and variance of the values in the $k$th dimension over $\tilde{\mathcal{U}}^a$. The smaller the Fisher discriminant ratio is, the less discriminative the two domains are. Theorem 2 holds with every $h \in \mathcal{H}$, so we omit PRBM here. $\mathtt{MR}\left(\bar{z}_i^g, \bar{z}^g\right)$ forces the local domain to be close to the global domain in terms of the mean value at each feature dimension in the feature representation independently, therefore, $\forall\, k \in [K]$,

$$\boldsymbol{\mu}_{\tilde{\mathcal{U}}_i^g}^k - \boldsymbol{\mu}_{\tilde{\mathcal{U}}^g}^k \leq \boldsymbol{\mu}_{\tilde{\mathcal{U}}_i}^k - \boldsymbol{\mu}_{\tilde{\mathcal{U}}}^k.$$

As the feature extractors share the same structure with identical parameter initialization and the feature representations are extracted from the same data domain $\mathcal{D}_i\,(\mathcal{D})$ [18, 34], we assume that $\boldsymbol{\sigma}_{\tilde{\mathcal{U}}_i^g} = \boldsymbol{\sigma}_{\tilde{\mathcal{U}}_i}$ and $\boldsymbol{\sigma}_{\tilde{\mathcal{U}}^g} = \boldsymbol{\sigma}_{\tilde{\mathcal{U}}}$. Thus, $\forall\, k \in [K]$,

$$\frac{\left(\boldsymbol{\mu}_{\tilde{\mathcal{U}}_i^g}^k - \boldsymbol{\mu}_{\tilde{\mathcal{U}}^g}^k\right)^2}{\left(\boldsymbol{\sigma}_{\tilde{\mathcal{U}}_i^g}^k\right)^2 + \left(\boldsymbol{\sigma}_{\tilde{\mathcal{U}}^g}^k\right)^2} \leq \frac{\left(\boldsymbol{\mu}_{\tilde{\mathcal{U}}_i}^k - \boldsymbol{\mu}_{\tilde{\mathcal{U}}}^k\right)^2}{\left(\boldsymbol{\sigma}_{\tilde{\mathcal{U}}_i}^k\right)^2 + \left(\boldsymbol{\sigma}_{\tilde{\mathcal{U}}}^k\right)^2}.$$

As this inequality is satisfied in all dimensions including the dimension where the maximum value exists, so for the Fisher discriminant ratio, we have

$$F1\left(\tilde{\mathcal{U}}_i^g,\tilde{\mathcal{U}}^g\right) = \max_k \left[ \frac{\left(\boldsymbol{\mu}_{\tilde{\mathcal{U}}_i^g}^k - \boldsymbol{\mu}_{\tilde{\mathcal{U}}^g}^k\right)^2}{\left(\boldsymbol{\sigma}_{\tilde{\mathcal{U}}_i^g}^k\right)^2 + \left(\boldsymbol{\sigma}_{\tilde{\mathcal{U}}^g}^k\right)^2} \right] \leq \max_k \left[ \frac{\left(\boldsymbol{\mu}_{\tilde{\mathcal{U}}_i}^k - \boldsymbol{\mu}_{\tilde{\mathcal{U}}}^k\right)^2}{\left(\boldsymbol{\sigma}_{\tilde{\mathcal{U}}_i}^k\right)^2 + \left(\boldsymbol{\sigma}_{\tilde{\mathcal{U}}}^k\right)^2} \right] = F1\left(\tilde{\mathcal{U}}_i,\tilde{\mathcal{U}}\right).$$

The smaller the Fisher discriminant ratio is, the less discriminative the two domains are. The less discriminative the two domains are, the smaller $d_{\mathcal{H}}\left(\cdot,\cdot\right)$ is. Thus, finally, we have

$$d_{\mathcal{H}}\left(\tilde{\mathcal{U}}_i^g,\tilde{\mathcal{U}}^g\right) \leq d_{\mathcal{H}}\left(\tilde{\mathcal{U}}_i,\tilde{\mathcal{U}}\right).$$

$\square$

## B.3 Derivations of Corollary 2

When we focus on the global-to-local knowledge transfer on the *client side*, in the FL scenario, we rewrite Theorem 1 as

**Theorem 3.** *Consider a virtual global data domain $\mathcal{D}$ and a local data domain $\mathcal{D}_i$. Let $\mathcal{D} = \langle \mathcal{U}, c^* \rangle$ and $\mathcal{D}_i = \langle \mathcal{U}_i, c^* \rangle$, where $\mathcal{U} \subseteq \mathcal{X}$ and $\mathcal{U}_i \subseteq \mathcal{X}$. Given a feature extraction function $\mathcal{F} : \mathcal{X} \mapsto \mathcal{Z}$ that shared between $\mathcal{D}$ and $\mathcal{D}_i$, a random labeled sample of size $m$ generated by applying $\mathcal{F}$ to a random sample from $\mathcal{U}$ labeled according to $c^*$, then for every $h \in \mathcal{H}$, with probability at least $1 - \delta$:*

$$\mathcal{L}_{\mathcal{D}_i}\left(h\right) \leq \mathcal{L}_{\hat{\mathcal{D}}}\left(h\right) + \sqrt{\frac{4}{m}\left(d\log\frac{2em}{d} + \log\frac{4}{\delta}\right)} + d_{\mathcal{H}}\left(\tilde{\mathcal{U}},\tilde{\mathcal{U}}_i\right) + \lambda_i,$$

*where $\tilde{\mathcal{U}}_i$ and $\tilde{\mathcal{U}}$ are the induced distributions of $\mathcal{U}_i$ and $\mathcal{U}$ under $\mathcal{F}$, respectively. $\tilde{\mathcal{U}}_i \subseteq \mathcal{Z}$ and $\tilde{\mathcal{U}} \subseteq \mathcal{Z}$. $\lambda_i := \min_h \mathcal{L}_{\mathcal{D}}\left(h\right) + \mathcal{L}_{\mathcal{D}_i}\left(h\right)$ denotes an oracle performance.*

**Corollary 2.** *Let $\mathcal{D}_i$, $\mathcal{D}$, $\mathcal{F}^g$, and $\lambda_i$ defined as in Corollary 1. Given a translation transformation function $\mathtt{PRBM} : \mathcal{Z} \mapsto \mathcal{Z}$ that shared between $\mathcal{D}_i$ and virtual $\mathcal{D}$, a random labeled sample of size $m$ generated by applying $\mathcal{F}'$ to a random sample from $\mathcal{U}_i$ labeled according to $c^*$, $\mathcal{F}' = \mathtt{PRBM} \circ \mathcal{F}^g : \mathcal{X} \mapsto \mathcal{Z}$, then for every $h' \in \mathcal{H}$, with probability at least $1 - \delta$:*

$$\mathcal{L}_{\mathcal{D}_i}\left(h'\right) \leq \mathcal{L}_{\hat{\mathcal{D}}}\left(h'\right) + \sqrt{\frac{4}{m}\left(d\log\frac{2em}{d} + \log\frac{4}{\delta}\right)} + d_{\mathcal{H}}\left(\tilde{\mathcal{U}}',\tilde{\mathcal{U}}_i'\right) + \lambda_i,$$

*where $d_{\mathcal{H}}\left(\tilde{\mathcal{U}}',\tilde{\mathcal{U}}_i'\right) = d_{\mathcal{H}}\left(\tilde{\mathcal{U}}^g,\tilde{\mathcal{U}}_i^g\right) \leq d_{\mathcal{H}}\left(\tilde{\mathcal{U}},\tilde{\mathcal{U}}_i\right) = d_{\mathcal{H}}\left(\tilde{\mathcal{U}}_i,\tilde{\mathcal{U}}\right)$. $\tilde{\mathcal{U}}'$ and $\tilde{\mathcal{U}}_i'$ are the induced distributions of $\mathcal{U}$ and $\mathcal{U}_i$ under $\mathcal{F}'$, respectively.*

*Proof.* PRBM is a translation transformation with parameters $\bar{z}_i^p$, *s.t.* $\forall\, \boldsymbol{x}_i \in \mathcal{U}_i, \boldsymbol{z}_i = \boldsymbol{z}_i^g + \bar{z}_i^p$, where $\boldsymbol{z}_i = \mathcal{F}'(\boldsymbol{x}_i) \in \tilde{\mathcal{U}}_i'$ and $\boldsymbol{z}_i^g = \mathcal{F}^g(\boldsymbol{x}_i) \in \tilde{\mathcal{U}}_i^g$. In other words, $\forall\, \boldsymbol{z}_i^g \in \tilde{\mathcal{U}}_i^g, \exists!\ \boldsymbol{z}_i \in \tilde{\mathcal{U}}_i'$. Therefore, we have $\mathrm{Pr}_{\tilde{\mathcal{U}}_i^g}[\{\boldsymbol{z} \in \mathcal{Z}\}] = \mathrm{Pr}_{\tilde{\mathcal{U}}_i'}[\{\boldsymbol{z} \in \mathcal{Z}\}]$ and the same applies to the pair of $\tilde{\mathcal{U}}^g$ and $\tilde{\mathcal{U}}'$, *i.e.*, $\mathrm{Pr}_{\tilde{\mathcal{U}}^g}[\{\boldsymbol{z} \in \mathcal{Z}\}] = \mathrm{Pr}_{\tilde{\mathcal{U}}'}[\{\boldsymbol{z} \in \mathcal{Z}\}]$. Then the subtraction of the probability on each side is also equal, *i.e.*,

$$\mathrm{Pr}_{\tilde{\mathcal{U}}_i^g}[\{\boldsymbol{z} \in \mathcal{Z}\}] - \mathrm{Pr}_{\tilde{\mathcal{U}}^g}[\{\boldsymbol{z} \in \mathcal{Z}\}] = \mathrm{Pr}_{\tilde{\mathcal{U}}_i'}[\{\boldsymbol{z} \in \mathcal{Z}\}] - \mathrm{Pr}_{\tilde{\mathcal{U}}'}[\{\boldsymbol{z} \in \mathcal{Z}\}].$$

$\forall\, h' \in \mathcal{H}, h^g = h' \circ \mathrm{PRBM} \in \mathcal{H}$, so $\forall\, \boldsymbol{z}^a \in \mathcal{Z}$ if $h^g(\boldsymbol{z}^a) = 1$, then $h'(\boldsymbol{z}^b) = 1$, where $\boldsymbol{z}^b = \boldsymbol{z}^a + \bar{z}_i^p$. Therefore, we have

$$\mathrm{Pr}_{\tilde{\mathcal{U}}_i^g}[\mathcal{Z}_{h^g}] - \mathrm{Pr}_{\tilde{\mathcal{U}}^g}[\mathcal{Z}_{h^g}] = \mathrm{Pr}_{\tilde{\mathcal{U}}_i'}[\mathcal{Z}_{h'}] - \mathrm{Pr}_{\tilde{\mathcal{U}}'}[\mathcal{Z}_{h'}],$$

where $\mathcal{Z}_{h^g} = \{\boldsymbol{z} \in \mathcal{Z} : h^g(\boldsymbol{z}) = 1\}, h^g \in \mathcal{H}$ and $\mathcal{Z}_{h'} = \{\boldsymbol{z} \in \mathcal{Z} : h'(\boldsymbol{z}) = 1\}, h' \in \mathcal{H}$. According to Definition 1, we have

$$\begin{aligned}
d_{\mathcal{H}}\left(\tilde{\mathcal{U}}', \tilde{\mathcal{U}}_i'\right) &= 2\sup_{h' \in \mathcal{H}}\left|\mathrm{Pr}_{\tilde{\mathcal{U}}_i'}[\mathcal{Z}_{h'}] - \mathrm{Pr}_{\tilde{\mathcal{U}}'}[\mathcal{Z}_{h'}]\right| \\
&= 2\sup_{h^g \in \mathcal{H}}\left|\mathrm{Pr}_{\tilde{\mathcal{U}}_i^g}[\mathcal{Z}_{h^g}] - \mathrm{Pr}_{\tilde{\mathcal{U}}^g}[\mathcal{Z}_{h^g}]\right| \\
&= d_{\mathcal{H}}\left(\tilde{\mathcal{U}}^g, \tilde{\mathcal{U}}_i^g\right) \\
&\leq d_{\mathcal{H}}\left(\tilde{\mathcal{U}}, \tilde{\mathcal{U}}_i\right).
\end{aligned}$$

$\square$

## C  Detailed Settings

### C.1  Implementation Details

We create the datasets for each client using six public datasets: Fashion-MNIST (FMNIST)[3], Cifar100[4], Tiny-ImageNet[5] (100K images with 200 labels) and AG News[6] (a news classification dataset with four labels, more than 30K samples per label). The MDL is calculated through the public code[7]. We run all experiments on a machine with two Intel Xeon Gold 6140 CPUs (36 cores), 128G memory, eight NVIDIA 2080 Ti GPUs, and CentOS 7.8.

### C.2  Hyperparameters of DBE

For hyperparameter tuning, we use grid search to find optimal hyperparameters, including $\kappa$ and $\mu$. Specifically, grid search is performed in the following search space:

- $\kappa$: 0, 0.001, 0.01, 0.1, 1, 5, 10, 20, 50, 100, 200, 500
- $\mu$: 0, 0.1, 0.2, 0.3, 0.4, 0.5, 0.6, 0.7, 0.8, 0.9, 1.0

In this paper, we set $\kappa = 50, \mu = 1.0$ for the 4-layer CNN, $\kappa = 1, \mu = 0.1$ for the ResNet-18, and $\kappa = 0.1, \mu = 1.0$ for the fastText. We only set different values for the hyperparameters $\kappa$ and $\mu$ on different model architectures but use identical settings for one architecture on all datasets. Different models exhibit diverse capabilities in both feature extraction and classification. Given that our proposed DBE operates by integrating itself into a specific model, it is crucial to tune the parameters $\kappa$ and $\mu$ to adapt to the feature extraction and classification abilities of different models.

As for the *criteria for hyperparameter tuning*, $\kappa$ and $\mu$ require different tunning methods according to their functions. Specifically, $\mu$ is a momentum introduced along with the widely-used moving average technology in approximating statistics, so for the model architectures that originally contain statistics collection operations (*e.g.*, the batch normalization layers in ResNet-18) one can set a relatively small value by tuning $\mu$ from 0 to 1 with a reasonable step size. For other model architectures, one can set a relatively large value for $\mu$ by tuning it from 1 to 0. The parameter $\kappa$ is utilized to regulate the magnitude of the MSE loss in MR. However, different architectures generate feature representations with varying magnitudes, leading to differences in the magnitude of the MSE loss. Thus, we tune $\kappa$ by aligning the magnitude of the MSE loss with the other loss term.

---

[3] https://pytorch.org/vision/stable/datasets.html#fmnist
[4] https://pytorch.org/vision/stable/datasets.html#cifar
[5] http://cs231n.stanford.edu/tiny-imagenet-200.zip
[6] https://pytorch.org/text/stable/datasets.html#ag-news
[7] https://github.com/willwhitney/reprieve

# D  Additional Experiments

## D.1  Convergence

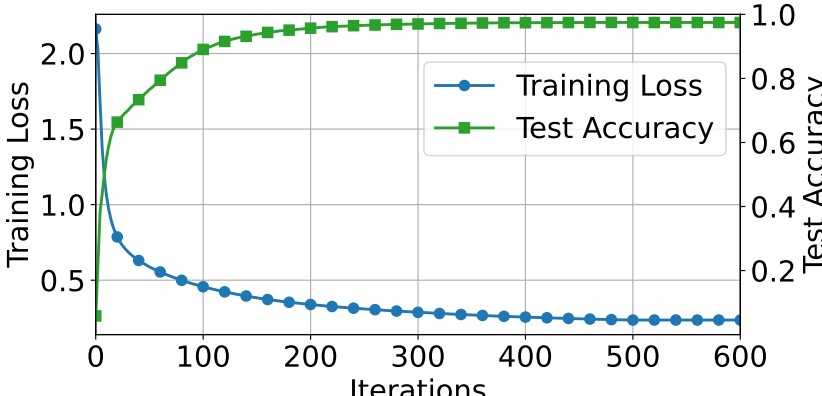

Figure 4: The training loss and test accuracy curve of FedAvg+DBE on FMNIST dataset using the 4-layer CNN in the practical setting.

Recall that our objective is

$$\min_{\boldsymbol{\theta}_1,\ldots,\boldsymbol{\theta}_N} \mathbb{E}_{i\in[N]}[\mathcal{L}_{\mathcal{D}_i}(\boldsymbol{\theta}_i)], \tag{11}$$

and its empirical version is $\min_{\boldsymbol{\theta}_1,\ldots,\boldsymbol{\theta}_N} \sum_{i=1}^{N} \frac{n_i}{n}\mathcal{L}_{\hat{\mathcal{D}}_i}(\boldsymbol{\theta}_i)$. Here, we visualize the value of $\sum_{i=1}^{N} \frac{n_i}{n}\mathcal{L}_{\hat{\mathcal{D}}_i}(\boldsymbol{\theta}_i)$ and the corresponding test accuracy during the FL process. Figure 4 shows the convergence of FedAvg+DBE and its stable training procedure. Besides, we also report the total iterations required for convergence on Tiny-ImageNet using ResNet-18 in Table 8. Based on the findings from Table 8, we observe that the utilization of DBE can yield a substantial reduction from 230 to 107 (more than 50%) in the total number of communication iterations needed for convergence, as compared to the original requirements of FedAvg.

## D.2  Model-Splitting in ResNet-18

In the main body, we have shown that DBE improves the per-layer MDL and accuracy of FedAvg no matter how we split the 4-layer CNN. In Table 7, we report the per-layer MDL and accuracy when we consider model splitting in ResNet-18, a model deeper than the 4-layer CNN. No matter at which layer, we split ResNet-18 to form a feature extractor and a classifier, DBE can also reduce MDL and improve accuracy, showing its general applicability.

Table 7: The MDL (bits, ↓) of layer-wise representations, test accuracy (%, ↑), and the number of trainable parameters (↓) in PRBM when adding DBE to FedAvg on Tiny-ImageNet using ResNet-18 in the practical setting. The "B", "CONV", "POOL", and "FC" means the "block", "convolution block", "average pool layer", and "fully connected layer" in ResNet-18 [29], respectively.

| Metrics | MDL | | | | | | | Accuracy | Param. |
|---|---|---|---|---|---|---|---|---|---|
| | CONV→B1 | B1→B2 | B2→B3 | B3→B4 | B4→POOL | POOL→FC | Logits | | |
| Original (FedAvg) | 4557 | 4198 | 3598 | 3501 | 3445 | 3560 | 3679 | 19.45 | 0 |
| CONV→DBE →B1 | 4332 | 4050 | 3528 | 3407 | 3292 | 3347 | 3493 | 19.96 | 16384 |
| B1→DBE →B2 | 4527 | 4072 | 3568 | 3456 | 3361 | 3451 | 3560 | 19.50 | 16384 |
| B2→DBE →B3 | 4442 | 4091 | 3575 | 3474 | 3326 | 3411 | 3520 | 19.55 | 8192 |
| B3→DBE →B4 | 4447 | 4073 | 3511 | 3414 | 3259 | 3346 | 3467 | 20.72 | 4096 |
| B4→DBE →POOL | 4424 | 4030 | 3391 | 3304 | 3284 | 3511 | 3612 | 39.99 | 2048 |
| POOL→DBE →FC | 4432 | 4035 | 3359 | 3298 | 3209 | 3454 | 3594 | 42.98 | 512 |

## D.3  Distinguishable Representations

As our primary goal is to demonstrate the elimination of representation bias rather than improving discrimination in Figure 3 (main body), we present the t-SNE visualization for our largest dataset in experiments, Tiny-ImageNet (200 labels). Given that the 200 labels are distributed around the chromatic circle, adjacent labels are assigned

similar colors, resulting in Figure 3 (main body) being indistinguishable by the label. Using a dataset AG News with only four labels for t-SNE visualization can clearly show that the representations extracted by the global feature extractor are distinguishable in Figure 5.

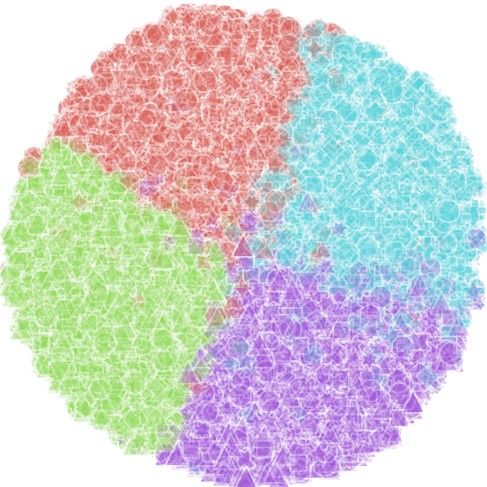

Figure 5: t-SNE visualization for the representations extracted by the global feature extractor on AG News (four labels) in FedAvg+DBE. We use *color* and *shape* to distinguish *labels* and *clients*, respectively.

## D.4 A Practical Scenario with New Participants

To simulate a practical scenario with new clients joining for future FL, we perform method-specific local training for 10 epochs on new participants for warming up after their local models are initialized by the learned global model (or client models in FedFomo). Since FedAvg, Per-FedAvg, and FedBABU do not generate personalized models during the FL process, we fine-tune the entire global model on new clients for them to obtain test accuracy. Specifically, using Cifar100 and 4-layer CNN, we conduct FL on 80 old clients ($\rho = 0.5$ or $\rho = 0.1$) and evaluate accuracy on 20 new joining clients after warming up. We utilize the data distribution depicted in Figure 9. According to Table 8, FedAvg shows excellent generalization ability with fine-tuning. However, DBE can still improve FedAvg by up to **+6.68** with more stable performance for different $\rho$.

Table 8: The total iterations for convergence and the averaged test accuracy (%, ↑) of pFL methods.

| Items | Iterations | New Participants | | Local Epochs | | |
|---|---|---|---|---|---|---|
| | | $\rho = 0.5$ | $\rho = 0.1$ | 1 | 5 | 10 |
| Per-FedAvg [22] | 34 | 48.66 | 48.36 | 95.10 | 93.92 | 93.91 |
| pFedMe [67] | 113 | 41.20 | 38.39 | 97.25 | 97.44 | 97.32 |
| Ditto [47] | 27 | 36.57 | 45.06 | 97.47 | 97.67 | 97.64 |
| FedPer [3] | 43 | 39.86 | 42.39 | 97.44 | 97.50 | 97.54 |
| FedRep [20] | 115 | 38.75 | 35.09 | 97.56 | 97.55 | 97.55 |
| FedRoD [14] | 50 | 50.10 | 51.73 | 97.52 | 97.49 | 97.35 |
| FedBABU [61] | 513 | 48.60 | 42.29 | 97.46 | 97.57 | 97.65 |
| APFL [21] | 57 | 38.19 | 45.16 | 97.25 | 97.31 | 97.34 |
| FedFomo [89] | 71 | 27.50 | 27.47 | 97.21 | 97.17 | 97.22 |
| APPLE [52] | 45 | — | — | 97.06 | 97.07 | 97.01 |
| FedAvg | 230 | 52.52 | 49.44 | 85.85 | 85.96 | 85.53 |
| FedAvg+DBE | 107 | **57.62** | **56.12** | **97.69** | **97.75** | **97.78** |

## D.5 Large Local Epochs

We also conduct experiments with more local epochs in each iteration on FMNIST using the 4-layer CNN, as shown in Table 8. All the pFL methods perform similarly with the results for one local epoch, except for Per-FedAvg, which degenerates around 1.18 in accuracy (%).

Table 9: The test accuracy (%) on the HAR dataset.

| Methods | Accuracy |
|---------|----------|
| FedAvg | 87.20±0.27 |
| SCAFFOLD | 91.34±0.43 |
| FedProx | 88.34±0.24 |
| MOON | 89.86±0.18 |
| FedGen | 90.82±0.21 |
| Per-FedAvg | 77.12±0.17 |
| pFedMe | 91.57±0.12 |
| Ditto | 91.53±0.09 |
| FedPer | 75.58±0.13 |
| FedRep | 80.44±0.42 |
| FedRoD | 89.91±0.23 |
| FedBABU | 87.12±0.31 |
| APFL | 92.18±0.51 |
| FedFomo | 63.39±0.48 |
| APPLE | 86.46±0.35 |
| FedAvg+DBE | **94.53±0.26** |

## D.6 Real-World Application

We also evaluate the performance of our DBE in a real-world application. Specifically, we apply DBE to the Internet-of-Things (IoT) scenario on a popular Human Activity Recognition (HAR) dataset [2] with the HAR-CNN [81] model. HAR contains the sensor signal data collected from 30 users who perform six activities (WALKING, WALKING_UPSTAIRS, WALKING_DOWNSTAIRS, SITTING, STANDING, LAYING) wearing a smartphone on the waist. We show the results in Table 9, where FedAvg+DBE still achieves superior performance.

## E  Broader Impacts

The representation bias and representation degeneration naturally exist in FL under statistically heterogeneous scenarios, which are derived from the inherently separated local data domains on individual clients. In the main body, we show the general applicability of our proposed DBE to representative FL methods. More than that, DBE can also be applied to other practical fields, such as the Internet of Things (IoT) [26, 31, 59] and digital health [15, 16]. Furthermore, introducing the view of knowledge transfer into FL sheds light on this field.

## F  Limitations

Although FL comes along for privacy-preserving and collaborative learning, it still suffers from privacy leakage issues with malicious clients [12, 93] or under attacks [23, 53]. We design DBE based on FL to improve generalization and personalization abilities, and we only modify the local training procedure without affecting the downloading, uploading, and aggregation processes. Thus, the DBE-equipped FL methods still suffer from the originally existing privacy issues like the original version of these FL methods when attacks happen. It requires future work to devise specific methods for privacy-preserving enhancement.

## G  Data Distribution Visualization

We illustrate the data distributions (including training and test data) in our experiments here.

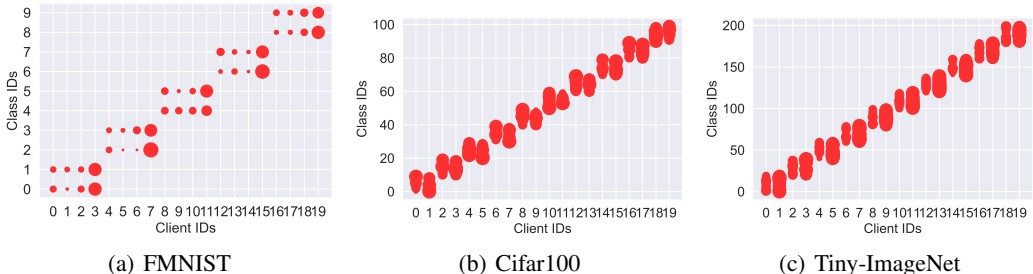

(a) FMNIST      (b) Cifar100      (c) Tiny-ImageNet

Figure 6: The data distributions of all clients on FMNIST, Cifar100, and Tiny-ImageNet, respectively, in the pathological settings. The size of a circle represents the number of samples.

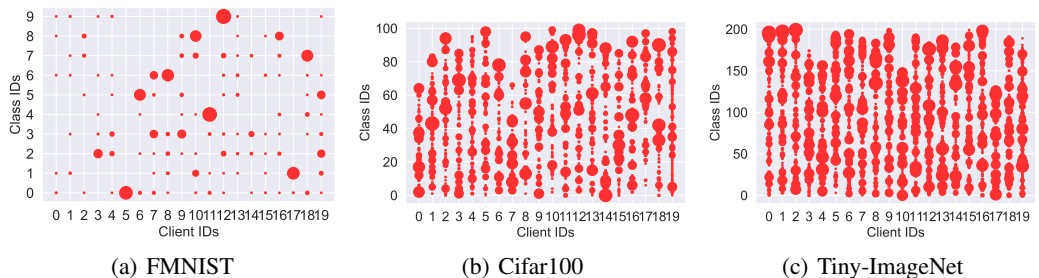

(a) FMNIST      (b) Cifar100      (c) Tiny-ImageNet

Figure 7: The data distributions of all clients on FMNIST, Cifar100, and Tiny-ImageNet, respectively, in the practical settings ($\beta = 0.1$). The size of a circle represents the number of samples.

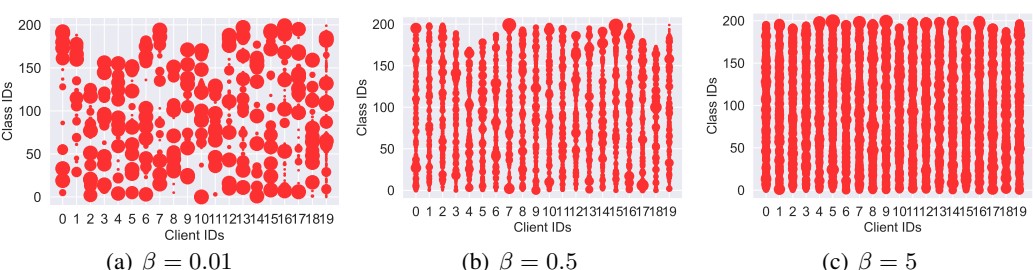

(a) $\beta = 0.01$      (b) $\beta = 0.5$      (c) $\beta = 5$

Figure 8: The data distribution on all clients on Tiny-ImageNet in three additional practical settings. The size of a circle represents the number of samples. The degree of heterogeneity decreases as $\beta$ in $Dir(\beta)$ increases.

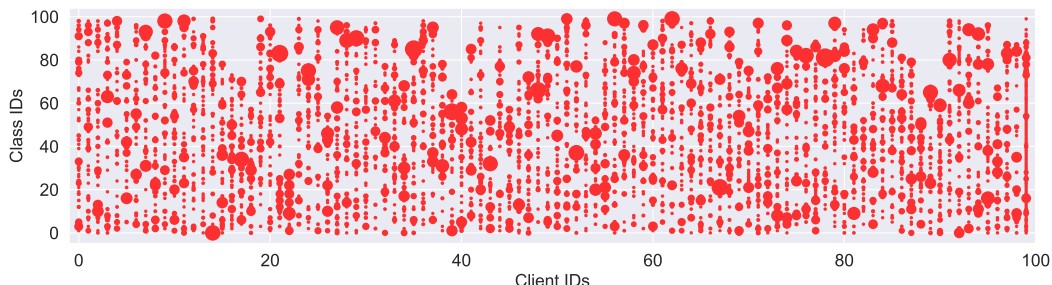

Figure 9: The data distributions of all clients on Cifar100 in the practical setting ($\beta = 0.1$) with 100 clients, respectively. The size of a circle represents the number of samples.

