# OpenReview forum: "Eliminating Domain Bias for Federated Learning in Representation Space"
_NeurIPS.cc/2023/Conference — NeurIPS 2023 poster_

### Official Review · Reviewer_xd4W · 2023-06-27

**Soundness:** 3 good
**Presentation:** 3 good
**Contribution:** 2 fair
**Rating:** 5
**Confidence:** 3

**Summary:**

This paper addresses the significant challenge of statistical heterogeneity among clients in federated learning. To tackle this issue, the authors propose personalized federated learning, where each client can learn and apply its specific model parameters. Specifically, the authors introduce DBE (Domain Bias Embedding), which comprises two crucial modules. The first module is a client-specific embedding designed to model the domain bias, while the second module is a regularization term incorporated into the objective function to encourage consensus in global representation learning. The effectiveness of these proposed modules is analyzed and demonstrated through theoretical results from domain generalization/adaptation. Moreover, extensive experiments are conducted to verify DBE's efficacy in reducing domain bias and its complementarity with other personalized federated learning methods.

**Strengths:**

1. The paper is well-written, effectively conveying the core idea in a clear and understandable manner.
2. The authors address the persistent issue of non-iidness in federated learning, which remains a significant challenge. The visualization of embeddings pre- versus post-local updates provides valuable motivation for studying and understanding this problem.
3. The proposed DBE (Domain Bias Embedding) approach is both simple and effective, demonstrating strong performance from both theoretical and empirical perspectives.
4. The paper's exploration of the complementary nature of DBE with existing personalized federated learning methods is particularly interesting. This aspect enhances the overall impact and relevance of the paper.

**Weaknesses:**

1. The introduction of the memory module requires improvement. While the module itself is relatively simple, the presentation of its implementation and functioning could be made clearer. It is challenging to understand the process and how to implement it based on the initial explanation.
2. The analysis of generalization risk, although provided, may have limited practical utility when it comes to the practical application of DBE. Further clarification or practical implications would enhance the relevance of this analysis.
3. The experimental evaluation appears to lack diversity in the considered tasks. Including evaluations on tasks such as graph learning would provide additional evidence and make the results more compelling.

**Questions:**

1. What's the difference between the numbers of layers that have memorized domain bias?
2. Is it possible to show that, in the iid setting, the memory embedding is zero?

**Limitations:**

None.

---

> ### Author Rebuttal · Authors · 2023-08-09
>
> Thank you for your time and constructive comments. We respond to your concerns as follows in the form of "**[A weakness or question]** Our responses".
>
>
> **[Improve the presentation of the memory module]** Thank you once again for your valuable suggestions. We will further refine the presentation of the memory module to enhance clarity.
>
>
> **[Practical application of $\texttt{DBE}$]** For the practical application, we apply our proposed $\texttt{DBE}$ to the IoT scenario on a popular Human Activity Recognition (HAR) dataset[1] with the HAR-CNN[2] model. HAR contains the sensor signal data collected from 30 users who perform six activities (WALKING, WALKING\_UPSTAIRS, WALKING\_DOWNSTAIRS, SITTING, STANDING, LAYING) wearing a smartphone on the waist. We show the results in R-xd4W-Table 1, where FedAvg+$\texttt{DBE}$ still achieves superior performance.
>
> R-xd4W-Table 1: The test accuracy (\%) on the HAR dataset.
> |  | Accuracy |
> |:-|:-:|
> | FedAvg |87.20±0.27 |
> | SCAFFOLD |91.34±0.43 |
> | FedProx |88.34±0.24 |
> | MOON |89.86±0.18 |
> | FedGen |90.82±0.21 |
> | Per-FedAvg |77.12±0.17 |
> | pFedMe |91.57±0.12 |
> | Ditto |91.53±0.09 |
> | FedPer |75.58±0.13 |
> | FedRep |80.44±0.42 |
> | FedRoD |89.91±0.23 |
> | FedBABU |87.12±0.31 |
> | APFL |92.18±0.51 |
> | FedFomo |63.39±0.48 |
> | APPLE |86.46±0.35 |
> | FedAvg+$\texttt{DBE}$ |**94.53±0.26** |
>
> [1] Anguita D, Ghio A, Oneto L, et al. Human activity recognition on smartphones using a multiclass hardware-friendly support vector machine. Ambient Assisted Living and Home Care: 4th International Workshop, IWAAL 2012.
>
> [2] Zeng M, Nguyen L T, Yu B, et al. Convolutional neural networks for human activity recognition using mobile sensors. 6th international conference on mobile computing, applications and services. IEEE, 2014: 197-205.
>
>
> **[Experimental evaluation lacks diversity]** In the FL field, most of the existing methods (e.g., SCAFFOLD, MOON, FedGen, Per-FedAvg, pFedMe, FedPer, FedRoD, FedBABU, APFL, FedFomo, and APPLE) primarily consider classification tasks within CV field, while we consider both CV and NLP fields. Graph learning is not a widely explored task in the traditional FL and pFL methods we have considered and compared. Nevertheless, our $\texttt{DBE}$ can eliminate domain bias in representation space for heterogeneous graphs on clients in theory, as the concept of representation is pervasive in deep learning tasks[3].
>
> [3] Bengio Y, Courville A, Vincent P. Representation learning: A review and new perspectives. IEEE transactions on pattern analysis and machine intelligence, 2013, 35(8): 1798-1828.
>
>
> **[What's the difference between the number of layers that have memorized domain bias?]** Our Personalized Representation Bias Memory ($\texttt{PRBM}$) module only contains a trainable vector $\bar{z}^p _i$ without any additional layers. If you are inquiring about the influence of the number of layers in the feature extractors under different model splitting methods, as discussed in *Section 5.1.1 How to split the model*, it is important to note that feature extractors with different numbers of layers possess varying feature extraction capabilities. According to previous studies [4,5], it has been observed that representations extracted by higher layers, which are farther away from the input, tend to exhibit a higher degree of bias compared to the ones extracted by lower layers.
>
> [4] Yosinski J, Clune J, Bengio Y, et al. How transferable are features in deep neural networks?. Advances in neural information processing systems, 2014.
>
> [5] Luo M, Chen F, Hu D, et al. No fear of heterogeneity: Classifier calibration for federated learning with non-iid data. Advances in Neural Information Processing Systems, 2021.
>
>
> **[Is it possible to show that, in the iid setting, the memory embedding is zero?]** While the impact of our $\texttt{PRBM}$ may be diminished in IID settings, its parameter vector $\bar{z}^p _i$ cannot be zero as long as there exist variations among clients in the representation space. Even in ideal IID settings, where each client trains models with the same dataset, the extracted representations of the same input on different clients often differ. This discrepancy arises from the stochastic optimization process employed by deep neural networks, such as stochastic gradient descent (SGD). Moreover, the use of different random seeds on different machines can further contribute to the variation in the extracted representations during each communication iteration. In practical FL scenarios, the data distribution among clients is often non-IID and statistically heterogeneous [6].
>
> [6] Li Q, Diao Y, Chen Q, et al. Federated learning on non-iid data silos: An experimental study. 2022 IEEE 38th International Conference on Data Engineering (ICDE). 2022.

---

> > ### Comment · Reviewer_xd4W · 2023-08-13
> > **Discussions**
> >
> > Thanks for your response! Most of my concerns are resolved. I also read the discussions with other reviewers. It seems that all reviewers are fond of this simple yet effective method. I will keep my score unchanged to support this paper.

---

> > > ### Author Response · Authors · 2023-08-14
> > > **Thank You!**
> > >
> > > Thank you once again for your timely feedback. We are grateful to have your support for our paper.

---

### Official Review · Reviewer_FLKZ · 2023-07-05

**Soundness:** 3 good
**Presentation:** 3 good
**Contribution:** 3 good
**Rating:** 6
**Confidence:** 4

**Summary:**

This paper tackles the problem of representation bias phenomenon lead by biased domains and proposes Domain Bias Eliminator (DBE), a framework that reduces domain discrepancy between server and client in representation space by Personalized Representation Bias Memory (PRBM) and Mean Regularization (MR). Experimental results show that DBE outperforming ten state-of-the-art methods in both generalization and personalization abilities.

**Strengths:**

-	Introduce the PRBM and MR to address the representation bias issue, and the effectiveness of these modules is validated by experimental results.
-	Provide theoretical analysis on generalization bounds of the global and local feature extractors.
-	This method has good compatibility and can be combined with other models.


**Weaknesses:**

-	It would be nice to evaluate these methods on real-world distribution shifts.
-	The introduction of MR increases the computational burden.
-	Some parts of the writing are not clear.


**Questions:**

1.	I'm confused about the form of $\bar{z}_i^p$, is it a trainable embedding vector (since it’s the personalized representation of client) or is it a model parameter (as Algorithm 1 claims)? Please elaborate on how $\bar{z}_i^p$ is derived.
2.	$\bar{z}^g$ is a consensus obtained during the initialization period, why is $\bar{z}^g$ not updated during the training process?
3.	What’s the dimension of the global $z_i^g$ and personalized $\bar{z}_i^p$ implemented in experiments?
4.	As mentioned in Line 249 “We run three trials for all methods until empirical convergence on each task”, does this mean that all models are trained until they converge and the communication rounds are different for each model (which is rare in federal learning settings)? And what are the specific conditions for convergence?
5.	Figure 1 in Appendix only contains the results of FedAvg+DBE, how about other methods? It would be better to compare the results with other methods to judge the convergence rate.
6.	The experimental results of the local model are missing in the paper
7.	In Section E.3, the author evaluated the accuracy on 20 new clients. How is the data of these clients divided, and does the class of these data appear in the 80 old clients?


**Limitations:**

The limitation of the proposed method has been discussed before.

---

> ### Author Rebuttal · Authors · 2023-08-09
>
> Thank you for your time and insightful feedback. We respond to your concerns as follows in the form of "**[A weakness or question]** Our responses".
>
>
> **[Evaluate methods on real-world distribution shifts]** Our $\texttt{DBE}$ is also effective in real-world scenarios. Here, we evaluate FedAvg+$\texttt{DBE}$ on the popular Human Activity Recognition (HAR) dataset[1] that contains the sensor signal data collected from 30 users who perform six activities wearing a smartphone on the waist. On HAR, following previous work[2], we use HAR-CNN as the model. We show the results in R-FLKZ-Table 1, where FedAvg+$\texttt{DBE}$ still achieves superior performance.
>
> R-FLKZ-Table 1: The test accuracy (\%) on the HAR dataset.
> |  | Accuracy |
> |:-|:-:|
> | FedAvg |87.20±0.27 |
> | SCAFFOLD |91.34±0.43 |
> | FedProx |88.34±0.24 |
> | MOON |89.86±0.18 |
> | FedGen |90.82±0.21 |
> | Per-FedAvg |77.12±0.17 |
> | pFedMe |91.57±0.12 |
> | Ditto |91.53±0.09 |
> | FedPer |75.58±0.13 |
> | FedRep |80.44±0.42 |
> | FedRoD |89.91±0.23 |
> | FedBABU |87.12±0.31 |
> | APFL |92.18±0.51 |
> | FedFomo |63.39±0.48 |
> | APPLE |86.46±0.35 |
> | FedAvg+$\texttt{DBE}$ |**94.53±0.26** |
>
> [1] Anguita D, Ghio A, Oneto L, et al. Human activity recognition on smartphones using a multiclass hardware-friendly support vector machine. Ambient Assisted Living and Home Care: 4th International Workshop, IWAAL 2012.
>
> [2] Zeng M, Nguyen L T, Yu B, et al. Convolutional neural networks for human activity recognition using mobile sensors. 6th international conference on mobile computing, applications and services. IEEE, 2014: 197-205.
>
>
> **[Computational burden brought by $\texttt{MR}$]** We discuss the additional computation overhead of our proposed $\texttt{MR}$ in *Section 4.5 Negligible Additional Communication and Computation Overhead*. The parameterless $\texttt{MR}$ computes the MSE loss for two representations of the $K$ dimension whose computational burden is negligible compared to model inference or backpropagation.
>
>
> **[Is the $\bar{z}^p _i$ a model parameter?]** Yes, as Algorithm 1 claims, $\bar{z}^p _i$ is a trainable model parameter in the client model when using $\texttt{DBE}$. As mentioned in *line 154*, $\bar{z}^p _i$ is the parameter of our personalized module $\texttt{PRBM}$, which is updated simultaneously with other parts of the client model. According to Figure 2, Equation (4), *line 155*, and Equation (8), we obtain representation ${z} _i$ via ${z} _i = {z}^g _i + \bar{z}^p _i$ during forward pass. For the backward pass, one can easily obtain the gradients of ${z} _i$, then the gradient of $\bar{z}^p _i$ can be derived by chain rule $\frac{\partial \mathcal{L} _{\hat{\mathcal{D}} _i}}{\partial \bar{z}^p _i} = \frac{\partial \mathcal{L} _{\hat{\mathcal{D}} _i}}{\partial {z} _i} \frac{\partial {z} _i}{\partial \bar{z}^p _i} = \frac{\partial \mathcal{L} _{\hat{\mathcal{D}} _i}}{\partial {z} _i}$.
>
>
> **[Why is ${z}^g$ not updated during the training process?]** We utilize ${z}^g$ to ensure consistent guidance for extracting client-invariant representation information in our $\texttt{MR}$. The $\texttt{MR}$ and $\texttt{PRBM}$ form a complementary pair. The updating of ${z}^g$ during the training process introduces dynamics to the guidance in $\texttt{MR}$, which can lead to a mismatch between the previously learned translation $\texttt{PRBM}$ and the current state of $\texttt{MR}$. Updating ${z}^g$ causes the test accuracy (\%) of FedAvg+$\texttt{DBE}$ to drop from 43.32 (Table 4, TINY, practical setting) to 41.55.
>
>
> **[The dimension of ${z}^g$ and $\bar{z}^p _i$ implemented in experiments]** The dimension of ${z}^g$ and $\bar{z}^p _i$ equals the dimension of feature representation space $\mathcal{Z}$, which depends on the model architectures and model splitting methods.
> Per *line 265-266*, we choose the last FC layer as the classifier, following existing methods, for a fair comparison. The dimensions $K$ are set to 512, 512, and 32 for the 4-layer CNN, ResNet-18, and fastText models, respectively, as indicated in our code.
>
>
> **[Specific conditions for convergence?]** If an FL method (algorithm) converges at the 100th round, running 100 rounds is equivalent to running 1000 rounds in terms of performance. Running all methods until empirical convergence is equivalent to running each method for a maximum required number of rounds, such as 1000 rounds, to ensure the empirical convergence of all methods. This approach is commonly used in FL experiments.
>
>
> **[Convergence rate of other methods]** We demonstrate Figure 1 in Appendix to show the convergence of FedAvg+$\texttt{DBE}$ rather than showing its superiority in convergence. Please note that we focus on improving the MDL and accuracy rather than emphasizing the superiority of the convergence rate. To compare the convergence rate, please refer to Table 5 (Overhead) to calculate the minimum communication iterations for convergence by dividing the values in the "Total time" column by the values in the "Time/iteration" column, i.e., Per-FedAvg: 34, pFedMe: 113, Ditto: 27, FedPer: 43, FedRep: 115, FedRoD: 50, FedBABU: 513, APFL: 57, FedFomo: 71, APPLE: 45, FedAvg: 230, FedAvg+$\texttt{DBE}$: 107. Using our $\texttt{DBE}$ reduces communication iterations for FedAvg. We will further improve this in the revised version.
>
>
> **[Results of the local model]** As stated in *line 244*, we adhere to the renowned pFedMe approach to present the results of the global model and personalized models for traditional FL and pFL methods, respectively. In pFL, the local model corresponds to the personalized model.
>
>
> **[The data distribution for *Section E.3* in Appendix]** We utilize the data distribution established in Table 4 (Cifar100†) consisting of 100 clients, as depicted in Figure 5 in the Appendix.

---

> > ### Comment · Reviewer_FLKZ · 2023-08-18
> >
> > Thank you for your detailed response and the additional experimental results. Most of the questions I raised have been addressed comprehensively. As you mentioned in the rebuttal, I hope to see a more detailed analysis of model convergence in the revised version. I will raise the score to 6 to support this work.

---

> > > ### Author Response · Authors · 2023-08-18
> > > **Thank You!**
> > >
> > > We are grateful for your valuable suggestions. Your support for our paper is greatly appreciated, and we will include a comprehensive analysis of model convergence in the revised version.

---

### Official Review · Reviewer_8bi2 · 2023-07-05

**Soundness:** 3 good
**Presentation:** 3 good
**Contribution:** 3 good
**Rating:** 7
**Confidence:** 4

**Summary:**

To address the performance drop via heterogeneous data distribution, the proposed method -  Domain Bias Estimator (DBE) - decomposes image features into unbiased (global) and biased (personalized) representations.

To guide the feature extractor to obtain the unbiased representation, Mean Regularization (MR) is proposed which is a regularization term for local objective functions that enforce to reduce the gap between mean of representations of local data and a mean of representation of all data across clients. Meanwhile, each client has learnable biased representation which is added to unbiased representation from the feature extractor, and then feed to the local classifier.

Unlike the previous works, by considering both unbiased and biased representation, the proposed method could improve bi-directional (local and global) knowledge transfer and could mitigate the performance drop via heterogeneity.

**Strengths:**

- The proposed method handles an important problem of non-IID FL.
- The paper is clearly written.
- The paper provides theoretical guarantees and comprehensive experimental results that show the efficiency of the proposed method for the given problem.
- The proposed method is novel in terms of decomposing representation generated by feature extractor into unbiased and biased representations, and efficient since it requires not much additional computational cost.

**Weaknesses:**

- The paper does not discuss about privacy issues that could be emerged by collection of client-specific mean over local data (line 2,3 in Algorithm 1, Supple). Since the client-specific mean contains representations from all local data, it is potentially exposed to reconstruct identifiable information of local data or even data itself.

**Questions:**

- Collecting averaged features extracted from all local data seems to be exposed to privacy attacks. It would be helpful to give more evidence (previous works or experiments) that the ‘client-specific mean’ collection is privacy-preserving.
- Figure 3 shows the role of client-variant (biased) representations and client-invariant (unbiased) representations. However, the representation is not separated according to labels. Are they distinguishable by label for representations extracted by the global feature extractor?

**Limitations:**

Yes, the authors well describe the limitation of the proposed framework.

---

> ### Author Rebuttal · Authors · 2023-08-09
>
> Thank you for your valuable feedback and insightful comments. We respond to your concerns as follows in the form of "**[A weakness or question]** Our responses".
>
>
> **[Privacy issues for client-specific mean collection (*line 2,3* in Algorithm 1, Supplementary)]** Please note that the client-specific mean $\{\bar{{z}}^g _1, \ldots, \bar{{z}}^g _N\}$ we collect during initialization period before FL is a $K$ dimensional vector for each client, respectively, instead of a set "containing representations from all local data". Hence, the client-specific mean solely captures the magnitude of the mean value for each feature dimension within the context of the given datasets and models. Sharing this kind of information is recently *popular* in the federated learning domain. FedPAC[1], FedProto[2], and FedPCL[3] share client-specific and class-wise mean (multiple $K$ dimensional vectors per client). Compared to them, the privacy-preserving ability of FedAvg+$\texttt{DBE}$ is superior, as they share both client-level and class-level information in each iteration while no class-level information is shared in FedAvg+$\texttt{DBE}$ and *we share the client-specific mean only once*. Furthermore, it is convenient to add privacy-preserving techniques (e.g., adding noise) to the client-specific mean during the initialization period without influencing the performance of our $\texttt{DBE}$, since the magnitude of the client-specific mean hardly changes with noise. Following FedPCL, we add Gaussian noise to the client-specific mean with controllable parameters scale ($s$) and perturbation coefficient ($p$). As shown in R-8bi2-Table 1, using the privacy-preserving technique can also bring slight improvement for our $\texttt{DBE}$.
> In the revised version of our paper, we will provide a more comprehensive privacy analysis to further strengthen it.
>
> R-8bi2-Table 1: The test accuracy (\%) on TINY in the practical setting.
> |  | Original | $s=0.05$, $p=0.1$ | $s=0.05$, $p=0.2$ |
> |:-|:-:|:-:|:-:|
> | FedAvg+$\texttt{DBE}$ | 43.32±0.12 | 43.81±0.15 | **44.10±0.10** |
>
> [1] Xu J, Tong X, Huang S L. Personalized Federated Learning with Feature Alignment and Classifier Collaboration. The Eleventh International Conference on Learning Representations. 2022.
>
> [2] Tan Y, Long G, Liu L, et al. Fedproto: Federated prototype learning across heterogeneous clients. Proceedings of the AAAI Conference on Artificial Intelligence. 2022.
>
> [3] Tan Y, Long G, Liu L, et al. Fedproto: Federated prototype learning across heterogeneous clients. Proceedings of the AAAI Conference on Artificial Intelligence. 2022.
>
>
> **[Are the representations extracted by the global feature extractor distinguishable by the label?]** Yes. As our primary goal is to demonstrate the elimination of representation bias rather than improving discrimination in Figure 3, we present the t-SNE visualization for our largest dataset in experiments, Tiny-ImageNet (200 labels). Given that the 200 labels are distributed around the chromatic circle, adjacent labels are assigned similar colors, resulting in Figure 3 being indistinguishable by the label. Using a dataset AG News with only four labels for t-SNE visualization can clearly show that the representations extracted by the global feature extractor are distinguishable in R-8bi2-Figure 1 (***please refer to the PDF in the global response field***).

---

> > ### Comment · Reviewer_8bi2 · 2023-08-12
> > **Thanks for the response**
> >
> > Thanks for the response and for providing additional representation visualization through t-SNE. Most of my concerns, including privacy issues, have been adequately addressed. I am particularly appreciative of their additional experiment using noisy representation (i.e. client-specific mean). It is intriguing that adding noise to client-specific mean achieves even better accuracy. This is noteworthy, especially considering that introducing noise to gradients typically causes a performance drop. Could the authors provide more insights into the phenomenon? I am wondering if it might be tied to the generalization ability, though I'm uncertain.

---

> > > ### Author Response · Authors · 2023-08-13
> > > **Thanks for New Comments**
> > >
> > > We appreciate your timely feedback and suggestions.
> > >
> > > First of all, we upload client-specific means and average them to generate the consensus vector, which is used to provide the unbiased feature information and preserve the feature magnitude for our $\texttt{MR}$. By sampling the noise from the same Gaussian distribution for all clients, the addition of moderate noise does not impact the magnitude of the consensus vector. Instead, it can be seen as incorporating additional unbiased information, which is beneficial for the representation bias elimination and can further improve the performance to some extent.
> > >
> > > Besides, it is not surprising that adding too much noise can also bring accuracy decrease, as shown in R-8bi2-Table 2 and R-8bi2-Table 3. "w/o" is short for "without". However, setting $s=0.05$ and $p=0.2$ is sufficient to ensure privacy protection according to FedPCL[3] (We apologize for the mis-citation in our previous response).
> > >
> > > R-8bi2-Table 2: The test accuracy (\%) on TINY in the practical setting with $s=0.05$ and larger $p$.
> > > |  | w/o noise | $p=0.1$ | $p=0.2$ | $p=0.5$ | $p=0.8$ | $p=0.9$ |
> > > |:-|:-:|:-:|:-:|:-:|:-:|:-:|
> > > | FedAvg+$\texttt{DBE}$ | 43.32±0.12 | 43.81±0.15 | 44.10±0.10 | 44.45±0.13 | 43.30±0.21 | 41.75±0.24 |
> > >
> > > R-8bi2-Table 3: The test accuracy (\%) on TINY in the practical setting with larger $s$ and $p=0.2$.
> > > |  | w/o noise | $s=0.05$ | $s=0.5$ | $s=1$ | $s=5$ |
> > > |:-|:-:|:-:|:-:|:-:|:-:|
> > > | FedAvg+$\texttt{DBE}$ | 43.32±0.12 | 44.10±0.10 | 44.15±0.18 | 43.78±0.14 | 36.27±0.35 |
> > >
> > >
> > > It is important to note that we upload the client-specific mean from clients to the server ***only once* before the FL process**. Our approach significantly differs from previous methods that upload averaged representations (such as class-specific prototypes) or model parameters ***per iteration* during the FL process**. Specifically,we add noise only once while previous methods continuously add noise throught the FL process. Therefore, they are significantly impacted by the noise.
> > >
> > >
> > > [3] Tan Y, Long G, Ma J, et al. Federated learning from pre-trained models: A contrastive learning approach. Advances in Neural Information Processing Systems, 2022.

---

> > > > ### Comment · Reviewer_8bi2 · 2023-08-15
> > > > **Thanks for the detailed response**
> > > >
> > > > The authors adequately addressed my major concern with additional experiments. I appreciate the timely response with proper experimental results and analysis. I found that there is no more significant drawback to the paper. Also, other peer reviewers thought that the method deserves acceptance. Thus, I raised my score to 7 to further support this paper. Thanks for the authors again and please include the comprehensive analysis on the privacy issue.

---

> > > > > ### Author Response · Authors · 2023-08-15
> > > > > **Thank You!**
> > > > >
> > > > > Thank you for your valuable feedback and insightful suggestions. We appreciate your strong support for our paper and will incorporate a thorough analysis of privacy concerns in the revised version.

---

### Official Review · Reviewer_dEaC · 2023-07-07

**Soundness:** 3 good
**Presentation:** 3 good
**Contribution:** 3 good
**Rating:** 7
**Confidence:** 3

**Summary:**

The paper introduces Domain Bias Eliminator (DBE) to address the representation bias and representation degeneration issues commonly observed in federated learning under statistically heterogeneous scenarios. DBE utilizes a Personalized Representation Bias Memory (PRBM) to preserve representation bias and mean regularization to guide local feature extractors toward producing representations with a consensual global mean. The authors provided a thorough theoretical analysis and empirical studies to validate their method. The results indicated that DBE can enhance the performance of existing FL methods.

**Strengths:**

1. This paper has solid theoretical analysis.
2. The proposed method is well-motivated.
3. Comparison with existing methods is sufficient.


**Weaknesses:**

1. The explanation on how DBE reduces the domain discrepancy between server and client in the representation space could be more detailed, as it forms a core part of the technique.

2. The authors could also consider exploring DBE's scalability to larger models, to understand the full potential of the method. The models used for experiments are all small, with only million-level parameters.

3. There are two main hyperparameters in the proposed method and it seems that the recommended choice is quite different for different datasets and models. It is difficult for practical scenarios to tune these hyperparameters.

**Questions:**

Is there any criteria or empirical methods to choose hyperparameters in this work?

**Limitations:**

The authors have addressed the limitations.

---

> ### Author Rebuttal · Authors · 2023-08-09
>
> Thank you for your constructive feedback and insightful comments. We respond to your concerns as follows in the form of "**[A weakness or question]** Our responses".
>
>
> **[More detailed explanation on how $\texttt{DBE}$ reduces the domain discrepancy]** As mentioned in *Section 4.4 Improved Bi-directional Knowledge Transfer*, our $\texttt{DBE}$ reduces $\mathcal{H}$-divergence (which measures domain discrepancy[1]) for both levels of representations. Here we show how $\texttt{DBE}$ reduces the domain discrepancy based on the widely used maximum mean discrepancy (MMD) metric[2]. $MMD[\Phi, P, Q] := \sup _{\phi \in \Phi}(\mathbb{E} _{p\sim P}[\phi(p)] - \mathbb{E} _{q\sim Q}[\phi(q)])$, where domains $P$ and $Q$ belong to a space $\mathcal{R}$ and $\Phi$ is a class of given functions $\phi: \mathcal{R} \rightarrow \mathbb{R}$. Our $\texttt{MR}$ is originally designed to reduce the mean discrepancy while our proposed translation $\texttt{PRBM}$ can keep the reduced mean discrepancy. Consider $P$ and $Q$ as the local representation domain and virtual global representation domain, respectively, and regard $\Phi$ as a class of identical summation function $\phi$, the MMD value can be reduced with the reduced mean discrepancy of $P$ and $Q$.
> We will continue to refine our paper in the revised version, aiming to enhance its impact and improve its clarity for better understanding.
>
> [1] Ben-David S, Blitzer J, Crammer K, et al. A theory of learning from different domains. Machine learning, 2010, 79: 151-175.
>
> [2] Gretton A, Borgwardt K, Rasch M, et al. A kernel method for the two-sample-problem. Advances in neural information processing systems, 2006.
>
>
> **[Small models with million-level parameters]** We follow prior approaches to adopt models for a fair comparison. The majority of existing FL approaches employ models with million (M)-level parameters. Specifically, shallow CNNs (around 5.6M parameters, used in FedAvg, FedGen, MOON, Ditto, FedRep, FedRoD, APFL, FedFomo, and APPLE) and ResNet-18 (around 11.2M parameters, used in FedBABU, FedALA[3], TCT[4], and ProgFed[5]) are popular models in the FL field.
>
> [3] Zhang J, Hua Y, Wang H, et al. Fedala: Adaptive local aggregation for personalized federated learning. Proceedings of the AAAI Conference on Artificial Intelligence. 2023.
>
> [4] Yu Y, Wei A, Karimireddy S P, et al. TCT: Convexifying federated learning using bootstrapped neural tangent kernels. Advances in Neural Information Processing Systems, 2022.
>
> [5] Wang H P, Stich S, He Y, et al. Progfed: effective, communication, and computation efficient federated learning by progressive training. International Conference on Machine Learning. 2022.
>
>
> **[Hyperparameter settings]** Please note that we **only** set different values for the hyperparameters $\kappa$ and $\mu$ on different model architectures but use identical settings for one architecture on all datasets. Different models exhibit diverse capabilities in both feature extraction and classification. Given that our proposed $\texttt{DBE}$ operates by integrating itself into a specific model, it is crucial to tune the parameters $\kappa$ and $\mu$ to adapt to the feature extraction and classification abilities of different models. As for the **criteria or empirical methods for hyperparameter tunning**, $\kappa$ and $\mu$ require different tunning methods according to their functions. Specifically, *$\mu$ is a momentum* introduced along with the widely-used moving average technology in approximating statistics, so for the model architectures that originally contain statistics collection operations (e.g., the batch normalization layers in ResNet-18) one can set a relatively small value by tuning $\mu$ from 0 to 1 with a reasonable step size. For other model architectures, one can set a relatively large value for $\mu$ by tuning it from 1 to 0. The parameter *$\kappa$ is utilized to regulate the magnitude of the MSE loss in our $\texttt{MR}$ (Equation (8))*. However, different architectures generate feature representations with varying magnitudes, leading to differences in the magnitude of the MSE loss. Thus, we tune $\kappa$ by aligning the magnitude of the MSE loss with the other loss term, i.e., $\frac{1}{n _i} \sum^{n _i} _{j=1} \ell(h(\texttt{PRBM}(f({x} _{ij}; {\theta}^f); \bar{z}^p _i); {\theta}^h), y _{ij})$.

---

### Author Rebuttal · Authors · 2023-08-09

We sincerely thank the reviewers for their valuable feedback on our manuscript. We have provided detailed responses in the rebuttal field following each reviewer’s comments. In this global response field, we provide a PDF file that includes a figure named R-8bi2-Figure 1.

---

### Decision · Program_Chairs · 2023-09-21

**Decision:**

Accept (poster)

**Comment:**

Initially, the reviewers were generally positive about the problem setting and the proposed method. They also appreciated the theoretical analysis and the thorough experimental evaluation. The author’s response clarified some aspects, and afterwards the reviewers did not see the need for further discussion. All reviewers recommend acceptance.